Manuscript prepared for Atmos. Chem. Phys.
with version 2015/11/06 7.99 Copernicus papers of the LaTeX class copernicus.cls.
Date: 16 July 2016

# Scalar turbulent behavior in the roughness sublayer of an Amazonian forest

Einara Zahn[1], Nelson L. Dias[2], Alessandro Araújo[3], Leonardo Sá[4],
Matthias Söergel[7], Ivonne Trebs[6], Stefan Wolff[7], and Antônio Manzi[8]

[1]Graduate Program in Environmental Engineering (PPGEA), Federal University of Paraná,
Curitiba, Brazil
[2]Department of Environmental Engineering, Federal University of Paraná, Curitiba, PR, Brazil
[3]Empresa Brasileira de Pesquisa Agropecuária (EMBRAPA), Trav. Dr. Enéas Pinheiro, Belém-PA,
CEP 66095-100, Brasil
[4]Centro Regional da Amazônia, Instituto Nacional de Pesquisas Espaciais (INPE), Belém, Pará,
Brazil
[6]Luxembourg Institute of Science and Technology, Environmental Research and Innovation (ERIN)
Department, L-4422 Belvaux, Luxembourg
[7]Biogeochemistry, Multiphase Chemistry, and Air Chemistry Departments, Max Planck Institute
for Chemistry, P.O. Box 3060, 55020, Mainz, Germany
[8]Instituto Nacional de Pesquisas da Amazônia (INPA), Clima e Ambiente (CLIAMB), Av. André
Araújo 2936, Manaus-AM, CEP 69083-000, Brazil

*Correspondence to:* Nelson L. Dias (`nldias@ufpr.br`)

**Abstract.** An important current problem in micrometeorology is the characterization of turbulence
in the roughness sublayer (RSL), where most of the measurements above tall forests are made. There,
scalar turbulent fluctuations display significant departures from the predictions of Monin–Obukhov
similarity theory (MOST). In this work, we analyze turbulence data of virtual temperature, carbon
dioxide and water vapor in the RSL above an Amazonian Forest (with a canopy height of 40 m),
measured at 39.4 and 81.6 m above the ground under unstable conditions. We found that dimensionless statistics related to the rate of dissipation of turbulence kinetic energy (TKE) and the scalar
variance display significant departures of MOST as expected, whereas the vertical velocity variance
follows MOST much more closely. Much better agreement between the dimensionless statistics with
the Obukhov similarity variable, however, was found for the subset of measurements made at low
zenith angle $Z$, in the range $0° < |Z| < 20°$. We conjecture that this improvement is due to the relationship between sunlight incidence and "activation/deactivation" of scalar sinks and fonts verticaly
distributed in the forest. Finally, we evaluated the relaxation coeficient of Relaxed Eddy Accumulation: it is also affected by zenith angle, with considerable improvement in the range $0° < |Z| < 20°$,
and its values fall within the range reported in the literature for the unstable Surface Layer. In general, our results indicate the possibility of better stability-derived flux estimates for low zenith angle
ranges.

# 1 Introduction

In the Atmospheric Surface Layer above the roughness sublayer (RSL) height $z_*$ (approximately three times the height of the roughness obstacles $h$ — Cellier and Brunet 1992), flux estimates based on mean concentration measurements are made with the help of Monin–Obukhov Similarity Theory (MOST) and the corresponding similarity functions. It is now well known, however, that MOST-based similarity functions often fail, to various degrees, in the roughness sublayer. In this region, beyond the classical governing variables found in MOST, there are several more intervening variables, such as tree spacing and vegetation density, among others (Garratt, 1980; Foken et al., 2012; Arnqvist and Bergström, 2015). This is the region where most mean concentration measurements above forests are made, and such is the case with the 82-m tower data analyzed in this work.

In principle, the availability of mean concentration data would make flux-gradient methods a natural choice to estimate scalar fluxes above the forest. Unfortunately, the difficulty of applying MOST in the RSL adds considerable uncertainty to this approach.

Maybe one of the earliest reports of the failure of flux-gradient methods when measurements are performed too close to the roughness elements was made by Thom et al. (1975), who compared flux-gradient and energy-budget Bowen ratio methods over a Pine forest, and found that the dimensionless gradients $\Phi_M$ and $\Phi_H$ of MOST are underestimated under such conditions. This was generally confirmed by Garratt (1978), who estimated $z_* = 4,5h$ for the momentum flux and $z_* = 3h$ for the sensible heat flux.

In the roughness sublayer, scalar and velocity gradients are weaker than above, leading to higher values of the corresponding turbulent diffusivities (Cellier and Brunet, 1992). Under neutral conditions, the momentum turbulent diffusivity increases by a factor 1.1–1.5, and the sensible heat turbulent diffusivity by 2–3. The turbulent Prandtl number correspondingly decreases from to close to 1 to approximately 0.5 at canopy top (Finnigan, 2000).

Cellier and Brunet (1992) propose a dimensionless factor $\gamma$ to account for the increase in turbulent diffusivity. Following this suggestion, Schween et al. (1997), using data measured over a 12-m tall oak and pine tree forest, found $\gamma_\theta = 2.2$. They pointed out that the different behavior in the RSL may be due to flux originating below the zero-plane displacement height, since in-canopy air may have significantly different characteristics from above-canopy air. As we will see, zenith angle analyses in the present work give support to this consideration.

Garratt (1980) proposed that for heights $z < z_*$ the vertical gradients depend on an additional length scale $z_s$, due to the turbulent wake generated by the trees. The author investigated data from a surface covered by scattered trees and shrubs in Australia, referred to as sub-tropical scrub or savannah, of average height 8 m and occupying about 25% of the total surface area. He analyzed data in the range $5 < d/z_0 < 85$ ($d$ is the zero-plane displacement height and $z_0$ is the surface roughness), suggesting an additional dependence of the dimensionless gradients on the variable $z/z_*$. Considering the canopy physical characteristics, he found that a relevant length scale is the tree spacing

$\delta$, and proposed $z_* = 3\delta$. Regarding Garratt's wake production assumption, it has been argued that this effect dies out rapidly above the canopy and that it is not the main cause for the lack of MOST-compliance in the roughness sublayer (Mölder et al., 1999).

Other attempts to organize roughness sublayer data include the use of $z/z_*$ by Cellier (1986), and Mölder et al. (1999)'s proposal of a function $(z/z_*)^n$ multiplying the dimensionless gradients:
Mölder et al. found $n = 1$ for scalars and $n = 0.6$ for momentum; they claim that the use of this factor produces acceptable results. Still, even with this correction, the resulting dimensionless functions in Mölder et al. (1999) display a much larger scatter than what is usually found above the roughness sublayer. Such roughness sublayer "dissimilarity" is not restricted to flux-gradient relationships: the dimensionless standard deviation of a scalar $a$, $\sigma_a/a_*$, has been found to be equally affected (Padro,
1993; Katul and Hsieh, 1999; von Randow et al., 2006; Williams et al., 2007; Dias et al., 2009).

In this paper, we analyze roughness-sublayer data collected under the scope of the ATTO project (Amazon Tall Tower Observatory), a German-Brazilian project undertaken under the leadership of Max-Planck-Institute (Germany), Instituto Nacional de Pesquisas da Amazônia (INPA) and Universidade Estadual do Amazonas (UEA) (Brazil). A 325-m tall tower has been erected in a forest
site 150 km NE of Manaus, and is currently undergoing instrumentation. Preliminary measurements have been made at an 82-m tall tower built at the site, and some analyses from the measured micrometeorological data are described here.

The main purpose of ATTO is to better understand the role of the Amazonian biome in the context of Global Climatic Changes. Specifically, the project aims at better understanding and modeling of
gaseous exchanges between the forest and the atmosphere (Andreae et al., 2015). For many scalars of interest, such as volatile organic compounds (VOCs), the high-frequency measurements needed in the Eddy Covariance Method are still difficult to make (particularly in long-term campaigns), leaving their flux estimates to methods based on the measurement of their mean concentrations and gradients.

Given the importance of correctly estimating trace gas fluxes over the Amazon forest, the lack of a theory for the roughness sublayer is clearly a major obstacle in the understanding of surface-atmosphere interactions with far-ranging implications on the regional and global hydrology, ecology, and climate.

Moreover, given the always present need to take into account site-specific features in any microm-
eteorological study, we attempt here to provide a general analysis of roughness sublayer-related questions at the ATTO site prior to the construction of the main tower. We expect that once measurements at the main tower become available, a better understanding of the questions preliminarily assessed here will be possible.

The variables analyzed in this study were chosen on the basis of data availability (the preliminary
campaigns had to be restricted to fewer variables than those that will be available once ATTO is fully implemented), as well as their usefulness to assess two main questions: (i) how does the Ama-

zonian RSL change the canonical (*i.e.*, measured above the RSL and reported as "classical") MOST similarity functions, and (ii) how useful/promising would flux-gradient and related methods applied within the Amazonian RSL be for the estimate of the fluxes of VOCs and other chemicals whose high-frequency measurement may be difficult to perform on a long-term basis?

In this work, the sections are organized as follows. In Sect. 2 we describe the experimental site in Amazonian Forest and data measurement; we also show the steps of data quality control. Theoretical concepts used to develop this research are reviewed in Sect. 3, where we describe the dissipation rate, vertical velocity skewness, spectral analysis and relaxed eddy accumulation in light of MOST. An important question of whether the RSL departures from MOST affect the inertial subrange behavior of the scalars is raised here, and an MOST function for the inertial subrange is proposed to address it. In Sect. 4 we discuss these results. Variance method results in the roughness sublayer and the zenith angle influence on several turbulence statistics are shown in Sect. 5, followed by an analysis of scalar similarity indices and their implications for flux estimation in Sect. 6. Finally, in Sect. 7 we make our final considerations.

## 2 Data measurement and quality control

### 2.1 Experimental site

The study area is located at Reserva de Desenvolvimento Sustentável Uatumã (RDSU) (Uatumã sustainable development reservation), in the counties of São Sebastião Uatumã and Itapiranga, in the Northeastern of Amazonas state, Brazil. The site is 150 km Northeast of the state capital Manaus, between the coordinates $59°10'$–$58°4'$W and $2°27'$–$2°4'$S.

In the forest, between 200 and 250 tree species per ha can be found, with a mean height of $40\,\mathrm{m}$ and with some individuals reaching 50 m. The site itself is located on a plateau (*terra firme*), with altitude 130 m.

The micrometeorological data were measured at an 82-m tower with a rectangular cross section of $2.5 \times 1\,\mathrm{m}^2$ at the site ($2°8'40''$S, $59°0'10''$W). Micrometeorological instrumentation was installed at the 23, 39.4 and 81.6 m levels (above ground).

In this work, we analyze pilot data from the 39.4 m and 81.6 m heights, measured during April, 2012. The data analyzed are the three wind components $u$, $v$ and $w$ measured by two sonic anemometers ( CSAT3, *Campbell Scientific Inc.* at 39.4 m; R3, *Gill Instruments Ltd.*, at 81.6 m), the sonic temperature (which we assume to be the same as the virtual temperature $\theta_v$), and the mass concentrations (mass of the species/total mass) of $CO_2$, $c$, and $H_2O$, $q$, calculated from the corresponding mass densities (mass of the species/volume) measured by two IRGA's (LI-7500A, *LI-COR Inc.*).

Both the CSAT3 and the Gill sonics report sonic virtual temperatures. The instantaneous values of $\theta_v$ from the sonics and of water vapor density $\rho_v$ and $CO_2$ density $\rho_c$ from the LI7500 were used to

derive instantaneous values of thermodynamic temperature $\theta$, of specific humidity $q$ and $CO_2$ mass concentration $c$. All our results for temperature were calculated with these corrected $\theta$ values.

In the same vein, the instantaneous fluctuations of $q$ and $c$ were used to calculate the fluxes and statistics of $H_2O$ and $CO_2$, without the need of further density corrections (Webb et al., 1980). This is sometimes called "direct method" (Miller et al., 2010; Prytherch, 2011), and produces results comparable to the WPL correction of Webb et al. (1980).

Recently, it has been found that the CSAT3 and Gill R3 sonics may require flow distortion corrections. For CSAT3, these corrections can change, *e.g.*, $w'$ estimates by approximately 8% Frank et al. (2013) and $\sigma_w$ by approximately 5–6% Frank et al. (2013); Horst et al. (2015). In our case, we tested the flow distortion corrections with the CSAT3 at the 39.4 m height, but our results changed very little: for example, the corrected $w'$ changed by slightly less than 4%. In this work, therefore, the data are processed without the aforementioned flow distortion corrections.

## 2.2 Quality control

The $10\,Hz$ data were analyzed in 30-min. data blocks ("runs"). Incomplete runs were excluded, and spikes were removed following Vickers and Mahrt (1997). For the next phase of quality control, fluctuations were extracted around a running mean. After that, each run was sub-divided into 15 two-minute subruns, and a local (i.e. 2-min.) standard deviation was calculated. Whenever this value was less than a threshold (pre-stipulated based on the sensor accuracy), the whole 30-min. run was excluded.

As a result, 21.5% of the 81.6-m level and 50.2% of the 39.4-m level runs were left. However, after this test, some strongly non-stationary time series remained, mainly in scalar data, even when linear detrending was applied. For this reason, these remaining data were further checked with two tests (both after removal of the linear trend). The first was the Reverse Arrangement Test (Bendat and Piersol 1986, p. 97; Dias et al. 2004), performed on $N = 50$ averages of the 30-min. run and a significance level $\alpha = 0.05$. The second test was defined by us based on a visual scrutiny of the data. It consisted of calculating the difference between the maximum and minimum values of the running mean within each 30 min. run. The run was discarded whenever this difference exceeded $\Delta\theta_v = 1.7$ $^\circ C$, $\Delta c = 0.11\,g\,kg^{-1}$ and $\Delta q = 3\,g\,kg^{-1}$. These further tests reduced data availability to 16.8% at the 81.6 m level and 41.5% at the 39.4-m level. For these runs, a 2-D coordinate rotation was applied for the final analyses.

As a percentage of the unstable runs only (which are the ones actually analysed in this work), the figures are 9.4% (81.6 m level) and 24.1% (39.4 m level). Although typical of micrometeorological studies, this somewhat low number of usable runs is bound to limit, for example, the percentage of reliable fluxes that can be retrieved in long-term studies. It is also likely that a larger number of runs fail quality control checks in the RSL in comparison with standard applications of MOST in the Surface Layer. In this work, we needed to concentrate on the analysis of good quality data

at the expense of time coverage. Parallel efforts will be required to increase data availablity and representativeness.

## 3  Theoretical background

In this section, we briefly review some results, which are used in the next section to analyze the data.

### 3.1  Dissipation rate of turbulence kinetic energy

The dimensionless dissipation rate of turbulence kinetic energy (TKE) is given by (Kaimal and Finnigan, 1994)

$$\phi_\epsilon = \frac{\kappa(z-d)\epsilon}{u_*^3}, \tag{1}$$

where $u_*$ is the friction velocity, $\kappa$ is the Von Karman constant and $\epsilon$ is the rate of dissipation of TKE. In MOST, a function still widely used to predict $\phi_\epsilon$ is (Kaimal et al., 1972)

$$\phi_\epsilon = (1 + 0.5|\zeta|^{2/3})^{3/2}, \quad -2 \le \zeta \le 0, \tag{2}$$

where $\zeta$ is the Monin–Obukhov stability parameter. As we shall see, in the roughness sublayer the dissipation rate deviates from the prediction by Eq. (2). In this work, we assess this depart from MOST by extending an index proposed by Mammarella et al. (2008):

$$\chi = \frac{u_*^3/\epsilon}{\kappa(z-d)/\phi_\epsilon} - 1 = \frac{L_\epsilon}{\kappa(z-d)} - 1 \tag{3}$$

In Eq. (1), $L_\epsilon$ is the length scale calculated from the friction velocity and the rate of dissipation of TKE, and adjusted to the effect of buoyancy: it can be regarded as an integral scale of the flow. It can readily be verified that $L_\epsilon/(\kappa(z-d)) = 1 \Rightarrow \chi = 0$ indicates that the dissipation data follow MOST perfectly. Originally, Mammarella et al. (2008) proposed $\chi$ to be used only under near-neutral conditions. As our data comprise too few near-neutral runs, it was necessary to take into account stability by including $\phi_\epsilon$ in the denominator of Eq. (1).

### 3.2  Vertical Velocity Skewness

In the surface layer, the skewnesses of $u$ and $w$ are

$$\mathrm{Sk}_w = \frac{\overline{w'^3}}{\sigma_w^3}, \tag{4}$$

$$\mathrm{Sk}_u = \frac{\overline{u'^3}}{\sigma_u^3}. \tag{5}$$

In convective or near-neutral conditions, $\mathrm{Sk}_w$ is typically observed to be negative in the roughness sublayer and positive in the inertial sublayer (Raupach and Thom, 1981; Fitzjarrald et al., 1990; von Randow et al., 2006). Some studies suggest that $\mathrm{Sk}_u$ tends to be positive in the roughness sublayer

and positive/near zero in the inertial sublayer (Kaimal and Finnigan, 1994; Kruijt et al., 2000), but there are also indications that it changes sign below or above the canopy height depending on vegetation density Poggi et al. (2004) . At some level above the canopy $Sk_w$ changes sign, and it seems reasonable to regard this level to be a measure of the roughness sublayer height. According to Fitz-jarrald et al. (1990), the negative $Sk_w$ values above canopies are largely due to the fact that there is

something below the "surface" [sic] in canopy layers, and there can be downward turbulent transport of vertical velocity variance associated with the drop in the TKE as one goes into the canopy. Kaimal and Finnigan (1994) attribute the considerable scatter in published results primarily to morphological differences between canopies, but at any rate this combination of strongly positive $u$-skewness and strongly negative $w$-skewness indicates that the turbulence is dominated by downward moving

gusts in the roughness sublayer.

### 3.3 Scalar dissipation from spectra and inertial subrange behavior

Consider the temperature spectrum in the inertial subrange, in the form

$$k^{5/3}E_{\theta\theta}(k) = \alpha_{\theta\theta}\epsilon^{-1/3}N, \tag{6}$$

where $\alpha_{\theta\theta} = 0.8$, and $N$ is the rate of dissipation of semi-temperature variance (Kaimal and Finni-

205 gan, 1994, p. 37).

Also on dimensional grounds, and under the validity of MOST, a similarity function exists that describes the temperature spectrum in the inertial subrange, viz.

$$g_\theta(\zeta) = \frac{\alpha_{\theta\theta}\epsilon^{-1/3}N(z-d)^{2/3}}{\theta_*^2}. \tag{7}$$

Again, we seek to determine to what degree $g_\theta$ calculated with roughness sublayer data obeys

MOST scaling. The usefulness of this indicator lies in its limited frequency-range: both $\epsilon$ and $N$ are inertial-subrange variables in the sense that they are obtained from a straightforward analysis of the inertial subrange of the velocity and temperature spectra. Therefore, $g_\theta$ is sensitive only to the highest range of frequencies (roughly $> 0.02\,\mathrm{Hz}$). If the dissimilarity displayed by "bulk" dimensionless statistics such as $\sigma_\theta/\theta_*$ is due to the larger scales only, then $g_\theta$ and similar variables should obey

MOST much more closely than the former. If on the other hand the dissimilarity is spread out through all frequency ranges, then $g_\theta$ should display the same sort of non-conformance to MOST as the other "bulk" statistics.

### 3.4 The Relaxed Eddy Accumulation method and related analyses

The original Eddy Acumulation method was proposed by Desjardins (1972) with the objective of es-

220 timating the turbulent flux of chemicals not easily measured at high frequency (see Ren et al., 2011). In the method, conditional sampling is performed on the gas, which is directed to either of two reservoirs according to the sign and intensity of the vertical wind velocity, $w$, by means of fast opening/closing valves. Evident advantages are the fact that the method dispenses with high-frequency

concentration measurements, and that only one-level measurements are needed (Tsai et al., 2012). Still, the method is not without difficulties, and to circumvent them Businger and Oncley (1990) proposed the simpler Relaxed Eddy Accumulation (REA) method, which uses the mean concentrations in each of two conditionally sampled reservoirs, $\overline{c^+}$ and $\overline{c^-}$, to calculate the flux as

$$\overline{w'c'} = b_c \sigma_w (\overline{c^+} - \overline{c^-}), \tag{8}$$

where $\sigma_w$ is the standard deviation of the vertical velocity and $b$ is the relaxation coefficient, that is empirically verified to be a dimensionless constant (it could be a MOST similarity function) of the order of 0.6 (Businger and Oncley, 1990; Katul et al., 1996).

Although initially developed and tested for the measurement of $CO_2$, water vapor and sensible heat fluxes (Pattey et al., 1993; Bash and Miller, 2007), the REA method is often extended for the measurement of other gaseous/scalar fluxes, usually with direct sensible or latent heat fluxes used as proxies to estimate $b$. For example, Zhu et al. (2000) used it to calculate ammonia fluxes, and Mochizuki et al. (2014) for isoprene and monotherpene fluxes; Matsuda et al. (2015) used it to estimate sulfate and $PM_{2.5}$ fluxes ; Bowling et al. (1998) used it to calculate isoprene fluxes, Bash and Miller (2007) and Sommar et al. (2013) for mercury fluxes and Moravek et al. (2014) used for peroxyacetyl nitrate fluxes.

In the ATTO project, we will be interested in the fluxes of several chemicals whose high-frequency measurement is still too laborious, too expensive, or downright impossible. Among these compounds are the VOCs released by the forests, the monoterpenes and isoprene being the most abundant followed by alcohols, carbonyls, acids, aldehydes, ketones and esters. If applicable, then, the REA method will be an invaluable tool.

However, strictly speaking, to be valid the method requires that the same value of $b$ apply to all scalars. A sufficient condition for this to happen, and the simplest — although by no means necessary — is the validity of the well-known hypothesis of perfect similarity between scalars (Hill, 1989; Dias and Brutsaert, 1996), which often fails under unstable conditions due to phenomena originating above the surface layer and to the transport of scalar variance from above (Cancelli et al., 2012, 2014). Furthermore, the physics of the roughness sublayer proper may complicate this picture even more.

Therefore, before applying the REA method to measurements made close to the canopy over a forest, it is important to assess both the validity of scalar similarity and the equality of the $b$'s for all scalars. A review of several REA studies by Tsai et al. (2012) showed that $b$ can vary as much as between 0.2 and 0.9, revealing the importance of its correct estimation. In Table 1 we give some values found in the literature outside of the roughness sublayer, and in Table 2 values found by Gao (1995) for the roughness sublayer for different values of the leaf-area index.

**Table 1.** REA coefficients measured above the roughness sublayer, where z is the measurement height (in meters) and numbers between brackets are the height of the canopies.

| Author(s) | Businger and Oncley (1990) | Katul et al. (1996) | Baker et al. (1992) |
|---|---|---|---|
| z(m) | 4 | 5 | – |
| Cover | crops | Corn (2.4 m) Grass (0.10 m) | soybean field |
| $b_\theta$ | 0.6 | $0.58 \pm 0.11$ | – |
| $b_q$ | 0.6 | $0.58 \pm 0.15$ | 0.56 |
| $b_c$ | – | $0.56 \pm 0.06$ | 0.56 |
| $b_{O_3}$ | – | $0.56 \pm 0.06$ | – |

**Table 2.** REA coefficients measured within the roughness sublayer (Gao, 1995).

| $z$ (m) | 43.1 | 34.2 | 18 | 14.4 | 10.5 | 5.9 |
|---|---|---|---|---|---|---|
| $b_\theta$ | $0.58 \pm 0.04$ | $0.55 \pm 0.04$ | $0.51 \pm 0.03$ | – | $0.61 \pm 0.06$ | $0.62 \pm 0.09$ |
| $b_q$ | – | $0.55 \pm 0.02$ | $0.51 \pm 0.05$ | $0.61 \pm 0.05$ | – | – |

## 4 Results

### 4.1 Dissipation rates

The rates of dissipation of TKE for each run were calculated from the longitudinal spectra on the basis of Kolmogorov's local isotropy theory (Kolmogorov, 1941). For each run, the inertial subrange was identified and a linear regression with an imposed $-5/3$ slope was calculated to determine $\epsilon$. With $\epsilon$, we calculated $\phi_\epsilon$ in Eq. (1) and the index $\chi$ in Eq. (1).

The results can be seen in Fig. 1 where the histograms of $\chi$ at two levels are shown. As mentioned 265 before, in the region of validity of MOST, we would expect the $\chi$'s to cluster around 0.

We find that, close to the canopy top, at 39.4 m, the integral scale $L_\epsilon$ often far exceeds $\kappa(z - d)$: this means that the latter is not a good estimate of the integral scale, as often found in the roughness sublayer. These results are in agreement with the findings by Rao et al. (1973) e Mammarella et al. (2008). At 81.6 m the spread of $\chi$ around zero is somewhat (but not very much) smaller, suggesting 270 that, as expected, we are reaching the upper limit of the roughness sublayer at these heights, again in agreement with what is usually found in the literature.

### 4.2 Velocity Skewness

In Fig. 2, we show the horizontal and vertical velocity skewness for the levels 39.4 m and 81.6 m, as a function of the stability parameter $\zeta$. The results confirm those found above with the dissipation rate

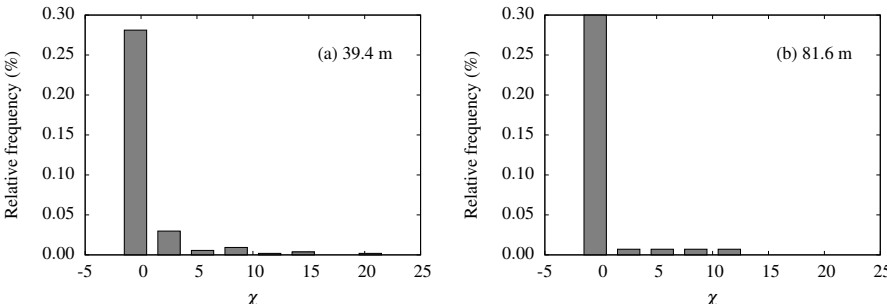

**Figure 1.** Histograms of $\chi$ (see Eq. ) at (a) 39.4 and (b) 81.6 cm.

in Sect. (4.1): roughness sublayer characteristics are still evident at the upper level, being obviously more pronounced at the lower level, where $Sk_w$ reaches close to $-0.8$. It is noteworthy that at the 39.4-m level, in spite of the prevalence of negative values of $Sk_w$, positive values typically associated with the surface layer can also be found. This suggests that the roughness sublayer height is actually varying from run to run, or that the physical picture is more complicated, with the possibility of positive skewnesses inside the roughness layer. The skewness $Sk_u$ of the longitudinal velocity shows the opposite behavior, with a predominance of positive values at the 39.4 m level, and a trend towards negative values higher up. This confirms, at least partially, the findings of Poggi et al. (2004). Clearly, the subject needs further research.

### 4.3 Inertial subrange similarity

Similarly to the analysis of the longitudinal velocity spectra, we identified for each run the inertial subrange of the temperature spectrum and fitted a linear regression with a $-5/3$ slope, in order to extract the rate of dissipation of semi-variance of temperature, $N$. From the latter, and $\epsilon$, the value of $g_\theta$ in Eq. (7) was calculated. It is plotted against $\zeta$ for the two levels, in Fig. 3-a and 3-b. There is a clear pattern of decreasing $g_\theta$ with $\zeta$, but still there is large scatter typical of roughness sublayer results.

This analysis suggests that inertial-subrange scales, approximately in the range 0.02–0.8 Hz, are also influenced by roughness sublayer effects: in other words, restricting the analysis to a range of smaller scales does not improve appreciably the predictions of MOST. Similar plots and results were also obtained for the other scalars (water vapor and $CO_2$), but are not shown here.

## 5 Variance method results

### 5.1 General behavior in the roughness sublayer

The "variance method", pioneered by Tillman (1972), is widely used in micrometeorology for several purposes, including quality control procedures (Foken and Wichura, 1996; Thomas and Foken,

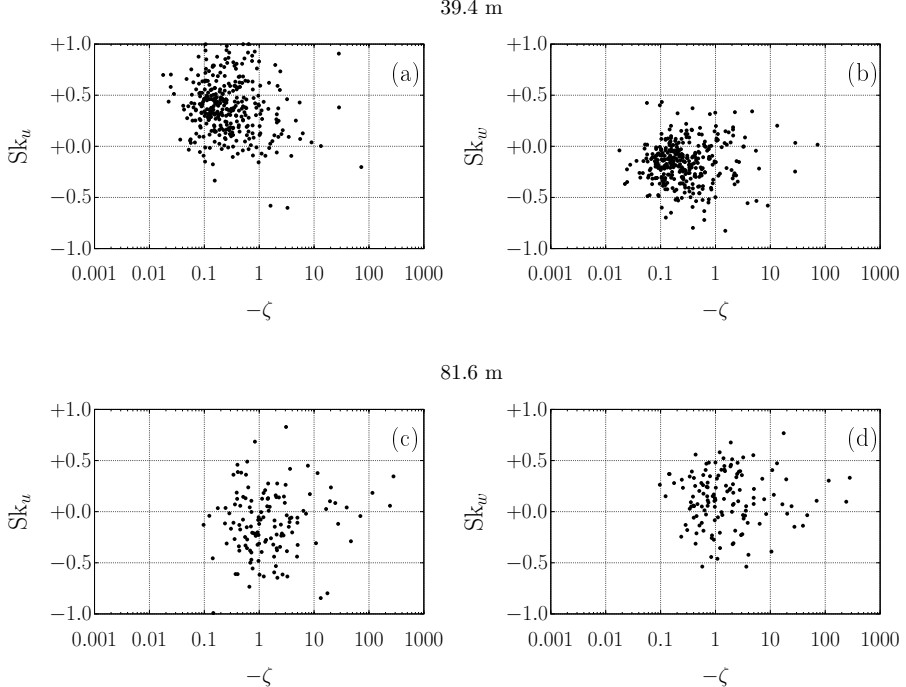

**Figure 2.** Horizontal (Sk$_u$) and vertical (Sk$_w$) velocity Skewness: (a) Sk$_u$ at 39.4 m; (b) Sk$_w$ at 39.4 m ; (c) Sk$_u$ at 81.6 m; and (d) Sk$_w$ at 81.6 m.

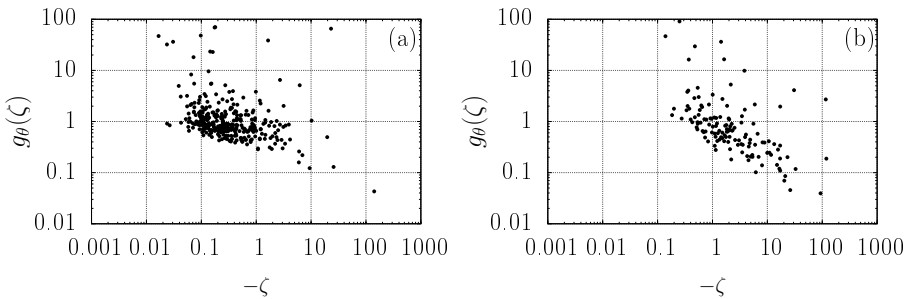

**Figure 3.** Inertial subrange similarity for temperature spectra: (a) 39.4 m and (b) 81.6 m.

2002; Lee et al., 2004) and flux estimation (Hong et al., 2008). Here, we consider similarity func-
300 tions of the type $\phi_a = \sigma_a/|a_*|$, where $\sigma_a$ is the scalar standard deviation, and $a_*$ the scalar's turbulent scale, defined by

$$\overline{w'a'} = u_* a_*, \tag{9}$$

as well as $\phi_w = \sigma_w/u_*$.

In the particular case of $\phi_w$ in stable conditions, it has been argued that its observed behavior for large $\zeta$ may be an artifact of the self-correlation effects raised by Hicks (1981) (see Pahlow et al. (2001)). Even though we are dealing with unstable conditions, we applied the randomization procedure recommended by Andreas and Hicks (2002) to test for possible self-correlation effects.

The randomized $\phi$-functions (not shown) tended to follow a $-1/3$ power law in the whole range of observed $\zeta$'s, even at near-neutral conditions, where the original data follow the predicted similarity functions. Our results are quite similar to those of Cava et al. (2008).

We also calculated the scalar fluxes for fairly to highly unstable conditions ($-\zeta > 0.2$) using the flux-variance method, and compared them with the measured fluxes. We obtained high correlations (in fact, higher than those reported by Cava et al. (2008)). As these authors comment, these fluxes are calculated without knowledge of $u_*$, and the success of the method under these more unstable conditions gives independent confirmation that spurious correlation is not contaminating our analyses.

Figures 4 and 5 show $\phi_w$, $\phi_c$, $\phi_q$ and $\phi_\theta$ for both measurement levels. Only negative $CO_2$ fluxes ($c_* < 0$) and positive latent and sensible heat fluxes ($q_* > 0$, $\theta_* > 0$) were considered, for unstable conditions $\zeta < 0$. In the figures, we plot empirical $\phi_a(\zeta)$ functions from experimental data for which good MOST agreement was observed (Katul et al., 1995).

Once more, the large scatter typical of roughness sublayer data is found: notice that the scatter is much larger in $\phi_\theta$, $\phi_c$ and $\phi_q$ than in $\phi_w$. Williams et al. (2007) suggest that this lack of MOST-compliance may be associated with heterogeneity of sources and sinks inside the canopy, contributing to the larger standard deviation (relative to the scalar turbulent scale).

This tendency of $\phi_a$ data in the roughness sublayer to lie above the corresponding MOST functions is generally observed in field experiments (Cava et al., 2008; Dias et al., 2009), but a definitive explanation for it is still lacking.

The $\phi_w$ data, on the other hand, show the oppostive trend (they fall *below* the corresponding MOST function for the surface sublayer), with much less scatter than in the scalar case.

Notice that this RSL "excess variance" (in comparison with MOST predictions) does not impact the fluxes (Katul et al., 1995); it does however make hypothetical flux estimates with the flux-variance method much more uncertain in the RSL (Dias et al., 2009). In the next section, we show this can be reduced signfcantly by taking into account the Zenith angle.

## 5.2   Zenith angle influence

It is known that the zenith angle ($Z$) can influence transfer characteristics between a vegetated surface and the atmosphere, e.g. the scalar roughness length (Sugita and Brutsaert, 1996). Iwata et al. (2010) note that, above tall vegetation, the vertical distribution of sources and sinks of scalars can vary both seasonally and during the day, depending on how deep light penetrates into the canopy.

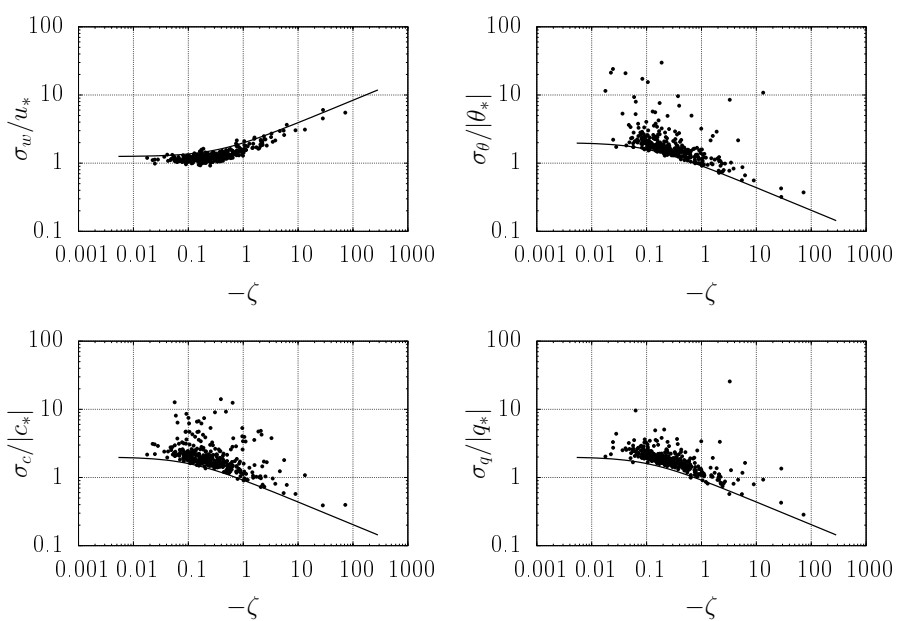

**Figure 4.** Dimensionless standard deviation for vertical velocity (a), temperature (b), $CO_2$ (c) and water vapor (d) — 39.4 m.

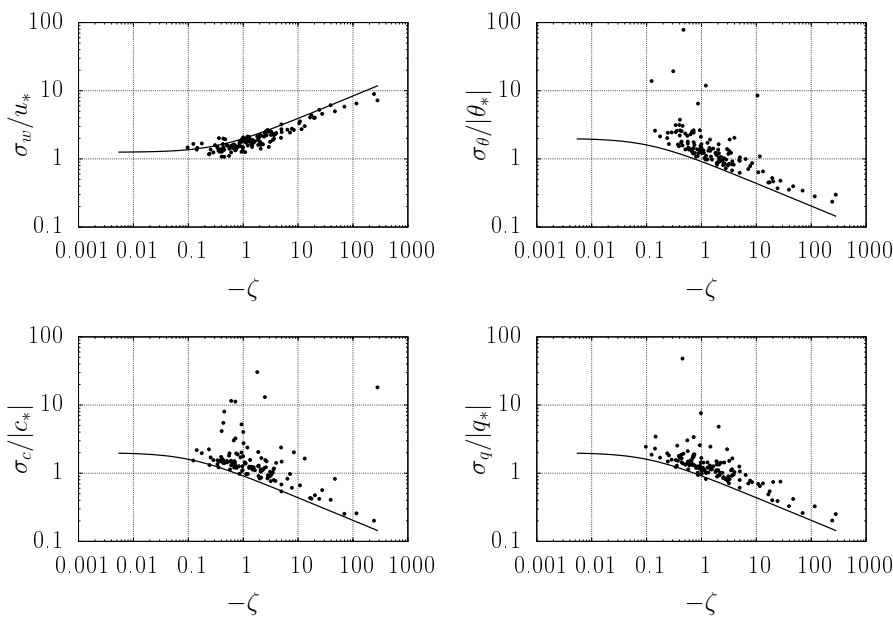

**Figure 5.** Dimensionless standard deviation for vertical velocity (a), temperature (b), $CO_2$ (c) and water vapor (d) — 81.6 m.

Therefore, the zenith angle is an obvious candidate as a possible effect on scalar transfer between

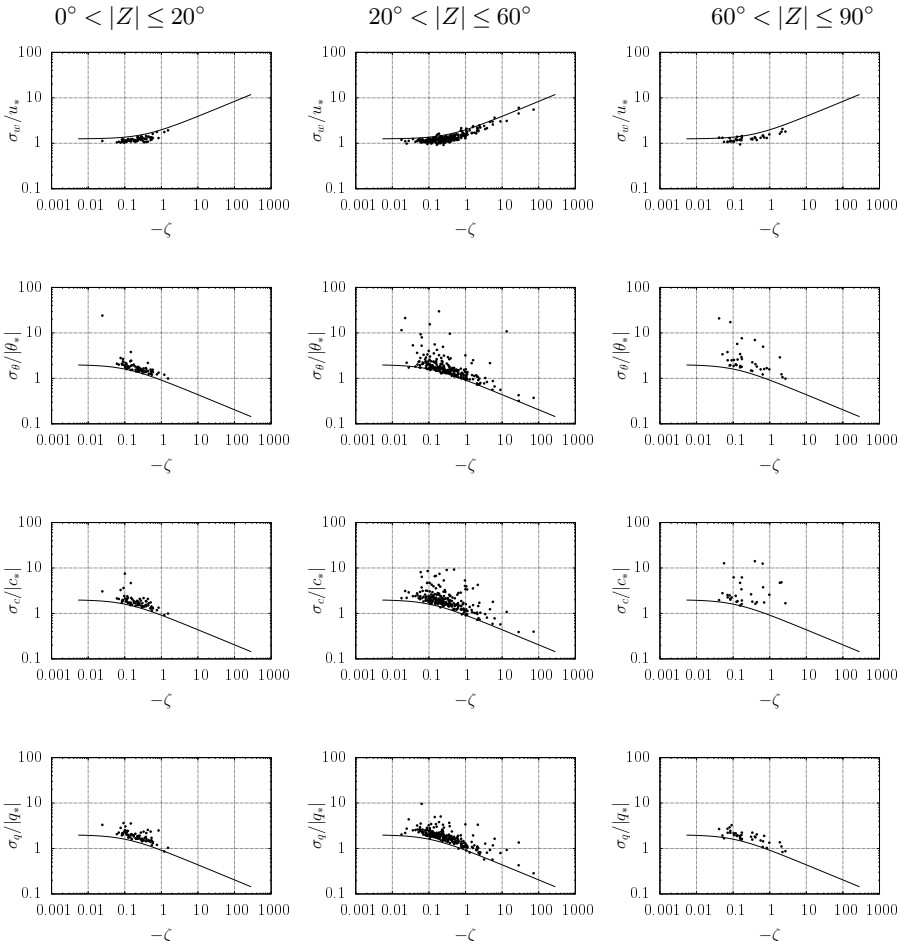

**Figure 6.** Dimensionless standard deviation with solar angle – 39.4 m. First row shows the velocity similarity function, second shows temperature function and third and fourth show $CO_2$ and water vapor, respectively.

the canopy and the atmosphere. In the following, we re-do the $\phi_a$ analysis for three different classes of zenith angle (which were found by trial-and-error to produce best results for the central class): $0° < |Z| \leq 20°$, $20° < |Z| \leq 60°$ and $60° < |Z| \leq 90°$.

In Fig. 6 (for the 39.4-m level), there is considerable improvement in the similarity relationships for all scalars when the sun is high, in the $0° < |Z| \leq 20°$ class of central zenith angles. Results are better for temperature than for $CO_2$ and $H_2O$, but this may be related to intrinsic difficulties of measuring the latter: dust and vapor condensation on the IRGA mirror surfaces, and the effects of sensor separation, are possible causes (Tsai et al., 2012). Similar results were found for the 81.6-m level.

These results are encouraging: at least in the hours around noon, similarity relationships as good as those observed in the surface layer over low vegetation can be obtained. This opens up the possiblity

of retrieving fluxes, by means of a host of standard MOST methods for these hours of the day. The results also require explanation. It is not immediately clear why these low zenith angles produce best results, but at least two (entirely phenomenological) explanations seem possible. One is vertical heterogeneity of sources and sinks, such as highlighted by Tsai et al. (2012): obviously, these sources and sinks may be more heterogeneous in the vertical under lower sunlight penetration. Horizontal heterogeneity, on the other hand, may also be playing a role: the vegetation height is not uniform, and a patchwork of shaded regions is clearly visible, from the tower, at higher values of the zenith angle. This may be enough to "activate"/"deactivate" sources of heat, $CO_2$ and $H_2O$ at the nominal source level $z - d$, producing what is effectively a non-homogeneous horizontal surface with local advection effects, which may be very hard to identify with standard techniques.

Still with respect to our results for the zenith angle, it is important to mention that similar effects were found by Detto et al. (2010) with regard to the storage term $z\frac{\partial \overline{s}}{\partial t}$, where $s$ is the scalar's concentration. They found that for small storage, of the order of 0.5% of the scalar flux, the scatter in the $\phi_{\theta,q,c,m}$ (where $m$ is for methane) was considerably smaller than under other conditions. Moreover, larger storages were observed around sunset and sunrise (notice that this will correspond to larger (in absolute value) zenith angles). Therefore, there may be a connection between Detto et al. (2010)'s results and ours. Unfortunately, concurrent good mean profile and turbulence data were not available in our dataset, precluding further analyses.

## 6  Scalar similarity indices and implications for flux estimation

### 6.1  Transport efficiencies

Not surprisingly, it turns out that sucess or failure of MOST-predictions in the roughness sublayer is also related to the degree of scalar similarity, although it seems that this aspect of RSL turbulence has not yet been fully explored. Two simple indices that are able to describe similarity between the fluxes of two scalars $a$ and $b$ are:

$$\text{rte}_{ab} = \frac{r_{wa}}{r_{wb}}, \tag{10}$$

$$\text{ste}_{ab} = 1 - \frac{||r_{wa}| - |r_{wb}||}{|r_{wa}| + |r_{wb}|}, \qquad 0 \leq \text{ste}_{ab} \leq 1, \tag{11}$$

where $r_{wa}$ is the correlation coefficient between the vertical velocity fluctuation $w'$ and the scalar fluctuation $a'$. Different from the correlation coefficient between the two scalars, $r_{ab}$, $\text{rte}_{ab}$ is a better descriptor of scalar flux similarity (Cancelli et al., 2012). $\text{ste}_{ab}$, proposed by Cancelli et al. (2012), has a similar interpretation, but is explicitly designed so that, unlike $\text{rte}_{ab}$, it is always bounded from above by 1.

Both $\text{rte}_{ab}$ and $\text{ste}_{ab}$ were calculated for all pairs of scalars. Next, we discuss the results for $39.4\,\text{m}$, shown in Fig. 7, 8 and 9 (the 81.6-m results are similar). Again, we find best scalar similarity for the $0° < |Z| \leq 20°$ range of zenith angles.

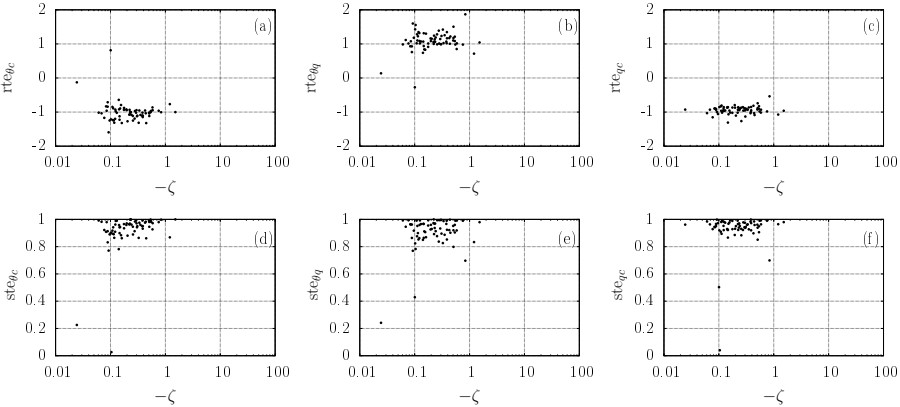

**Figure 7.** Scalar flux similarity indices rte (above) and ste (below) for $0° < |Z| \leq 20°$; $\theta$–$c$ (a), (d); $\theta$–$q$ (b), (e); and $q$–$c$ (c), (f). For rte, some data points were off the scale.

For most of the runs in the range $0° < |Z| \leq 20°$, ste is between 0.8 and 1.0. The most similar *pair* of scalar fluxes was $CO_2$–temperature (c.f. Fig. 7-a and 7-d). We conjecture that this is related to the fact that both scalars have the oposite sign of entraiment fluxes at the top of the atmospheric boundary layer and at the ground; this implies that they are anticorrelated in all the extension of the Atmospheric Boundary Layer (ABL). This is quite common in the unstable ABL. This behaviour is not verified for the $H_2O$–temperature and $CO_2$–$H_2O$ pairs: these pairs of scalar fluxes are correlated ($H_2O$–temperature) and anticorrelated ($CO_2$–$H_2O$) at the ground and are anticorrelated ($H_2O$–temperature) and correlated ($CO_2$–$H_2O$) at the top of the ABL. This relationship between entrainment flux and similarity indices was observed, for example, in the LES simulations of Cancelli et al. (2014). However, it is clear that both roughness sublayer and entrainment effects may be impacting scalar similarity, and in that regard more research is needed. For instance, Detto et al. (2008) suggest that these two effects can be discerned by quadrant analysis.

The similarity indices for the other zenith angle intervals are shown in Fig. 8 and 9. In the interval $20° < |Z| \leq 60°$, the scalar fluxes still display a certain degree of similarity. In the interval $60° < |Z| \leq 90°$, however, little similarity is observed. We note that these data often correspond to early morning and late afternoon periods, when scalar sources are usually "deactivated". These results reinforce the picture of scalar fluxes emanating/being absorbed from different sources/sinks within the canopy and in the soil (c.f. Scanlon and Kustas (2010)).

### 6.2 Relaxed Eddy Accumulation

As reviewed in Sect. 3, the relaxed eddy accumulation is a valuable alternative for the measurement of scalars for which fast-response sensors are not available. This comes at the cost of the extra hypothesis of perfect scalar similarity.

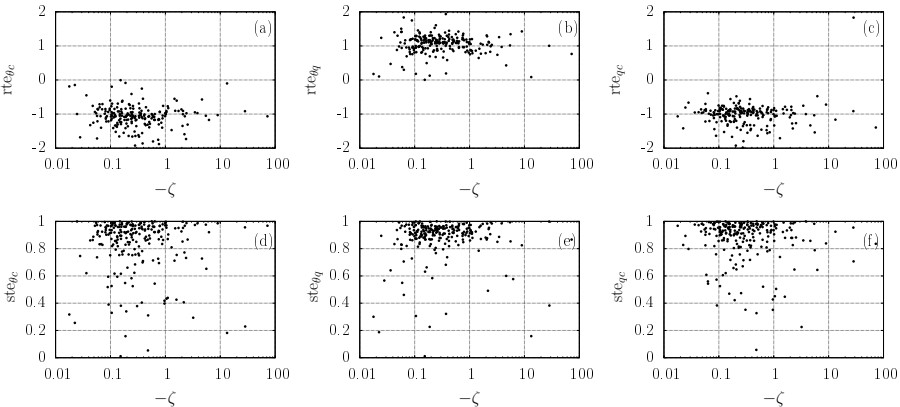

**Figure 8.** Scalar flux similarity indices rte (above) and ste (below) for $20° < |Z| \leq 60°$; $\theta$–$c$ (a), (d); $\theta$–$q$ (b), (e); and $q$–$c$ (c), (f). For rte, some data points were off the scale.

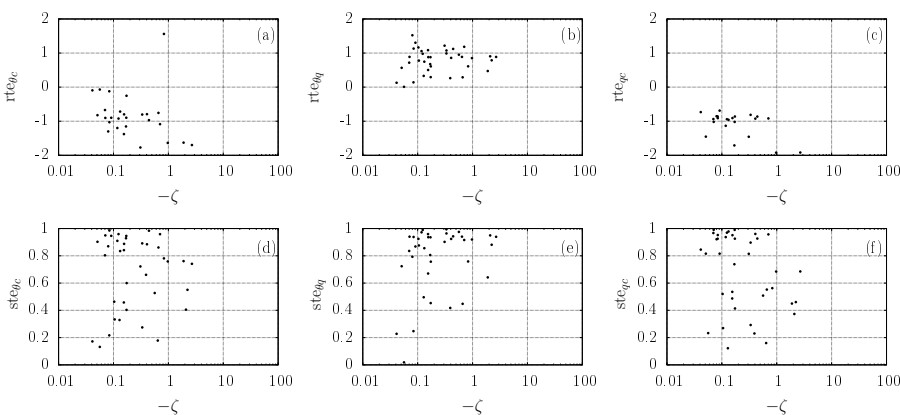

**Figure 9.** Scalar flux similarity indices rte (above) and ste (below) for $60° < |Z| \leq 90°$; $\theta$–$c$ (a), (d); $\theta$–$q$ (b), (e); and $q$–$c$ (c), (f). For rte, some data points were off the scale.

Since there was no REA system installed at the tower, we have simulated the method using the eddy-covariance data. The fast scalar data were used to obtain updraft and downdraft REA samples by conditional sampling of $\theta$, $c$ and $q$ values for $w > 0$ and $w < 0$, respectively. These simulated samples were then averaged to obtain $\overline{\theta^+}$, $\overline{\theta^-}$, $\overline{c^+}$, $\overline{c^-}$, $\overline{q^+}$ and $\overline{q^-}$.

Given the results found in the previous subsection for rte and ste, it is natural to expect the REA method to perform better, again, in the range $0° < |Z| \leq 20°$. In the following we analyze the relaxation coefficients $b$ defined in Eq. (8), according to scalar and zenith angle.

The overall results, not classified according to zenith angle, are shown in Fig. 10, again for unstable conditions only. A few cases for which $b < 0$ were discarded, since the REA, similarly to

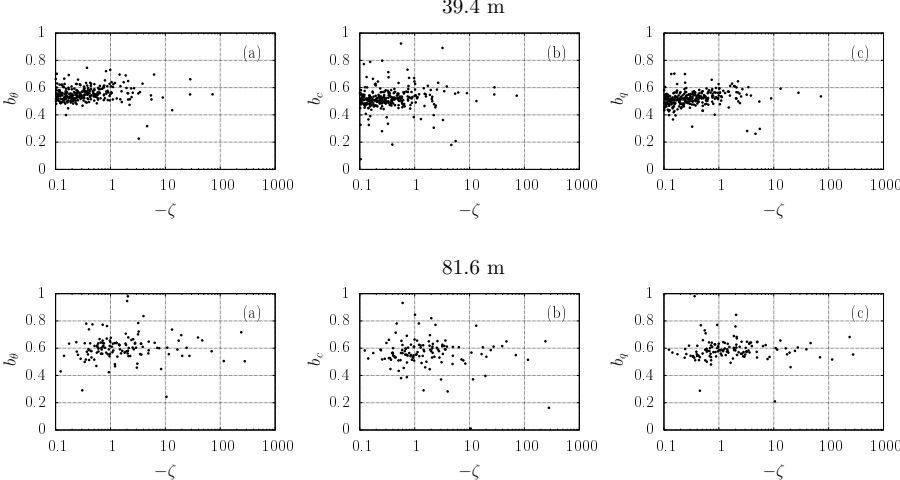

**Figure 10.** Relaxation coefficient from Eddy Accumulation Method, where (a) and (d) are temperature, (b) and (e) are $CO_2$ and (c) and (f) are water vapor.

**Table 3.** Statistics from REA in 39.4 and 81.6 m, where $b_\theta$, $b_c$ and $b_q$ are the relaxation coeficients for temperature, $CO_2$ and water vapor, respectively.

|  | 39.4 m | | | 81.6 m | | |
| --- | --- | --- | --- | --- | --- | --- |
|  | $b_\theta$ | $b_c$ | $b_q$ | $b_\theta$ | $b_c$ | $b_q$ |
| Mean | 0.560 | 0.535 | 0.522 | 0.625 | 0.634 | 0.593 |
| Median | 0.550 | 0.517 | 0.519 | 0.603 | 0.567 | 0.584 |
| Standard deviation | 0.091 | 0.158 | 0.075 | 0.215 | 0.710 | 0.095 |

flux-gradient methods, cannot cope with counter-gradient fluxes. Our data does not show any significant dependency of $b$ on $\zeta$, in agreement with Katul et al. (1996) and Businger and Oncley (1990).

In table 3, we give the (overall) means, medians and standard deviations of $b$ for each scalar. All the means are in the same range found by other authors (e.g. $0.51 - 0.62$ by Katul et al. (1996)), which incidentally are values obtained for the surface layer *above* the roughness sublayer. The scatter in our data, however, is larger: at 81.6 m it reaches 0.71 for $b_c$.

Our mean values of $b$ are also somewhat smaller at 39.4 m (between 0.52 and 0.56) than at 81.6 m (between 0.59 and 0.63). This is similar to the results of Gao (1995) (shown in Table 2), where the mean $b_{\theta,q}$ values are 0.51 next to the canopy top (18 m) and 0.58 in 43.1 m (approximately two times the mean forest height average).

**Table 4.** Statistics from REA for each zenith angle class

| | 39.4 m | | | 81.6 m | | |
|---|---|---|---|---|---|---|
| $0° < |Z| \leq 20°$ | $b_\theta$ | $b_c$ | $b_q$ | $b_\theta$ | $b_c$ | $b_q$ |
| Mean | 0.555 | 0.505 | 0.505 | 0.613 | 0.588 | 0.587 |
| Median | 0.546 | 0.510 | 0.517 | 0.622 | 0.594 | 0.589 |
| Standard deviation | 0.039 | 0.076 | 0.047 | 0.071 | 0.064 | 0.091 |
| $20° < |Z| \leq 60°$ | $b_\theta$ | $b_c$ | $b_q$ | $b_\theta$ | $b_c$ | $b_q$ |
| Mean | 0.558 | 0.548 | 0.526 | 0.632 | 0.665 | 0.597 |
| Median | 0.551 | 0.521 | 0.518 | 0.596 | 0.557 | 0.582 |
| Standard deviation | 0.055 | 0.176 | 0.084 | 0.252 | 0.871 | 0.087 |
| $60° < |Z| \leq 90°$ | $b_\theta$ | $b_c$ | $b_q$ | $b_\theta$ | $b_c$ | $b_q$ |
| Mean | 0.591 | 0.522 | 0.537 | 0.731 | 0.582 | 0.654 |
| Median | 0.546 | 0.500 | 0.532 | 0.772 | 0.551 | 0.620 |
| Standard deviation | 0.225 | 0.147 | 0.044 | 0.138 | 0.189 | 0.094 |

We classify the $b$-values according to zenith angle in Table 4. For each zenith angle interval, means and medians are quite similar among the three scalars. Moreover, for most cases in the intervals $0° < |Z| \leq 20°$ and $20° < |Z| \leq 60°$ the standard deviations are larger at 81.6 m.

For $0° < |Z| \leq 20°$, the standard deviations of $b_\theta$, $b_c$ and $b_q$ are significantly smaller (by a factor of 2) than for the other two classes, as well as for all data considered together. We note that in general the standard deviations for $60° < |Z| \leq 90°$ are smaller than those for $20° < |Z| \leq 60°$, but they are also more uncertain, due to the small number of points falling into that class. Overall, the better results for small zenith angles are confirmed for the REA method.

A version of the REA for momentum is possible. Although it is seldom used to estimate $u_*$ (see Andreas et al. (1998) for such an application), it reads

$$\overline{w'u'} = b_u \sigma_w (\overline{u^+} - \overline{u^-}). \tag{12}$$

To best of our knowledge, the behavior of $b_u$ in the roughness sublayer has not been examined. In table 5 we show the $b_u$ values for the whole data at 39.4 m, as well as for each zenith angle class. As already observed for the scalars, and not shown here, the $b_u$ behavior at 81.6 m is similar, although somewhat more scattered. The $b_u$ behavior mimics that of the scalars in the sense that its means and medians do not vary significantly with the zenith angle, but the scatter (measured by its standard deviation) is much larger for the second class (again the the last class has too few points, which may explain its smaller standard deviation). Therefore, although the zenith angle does not

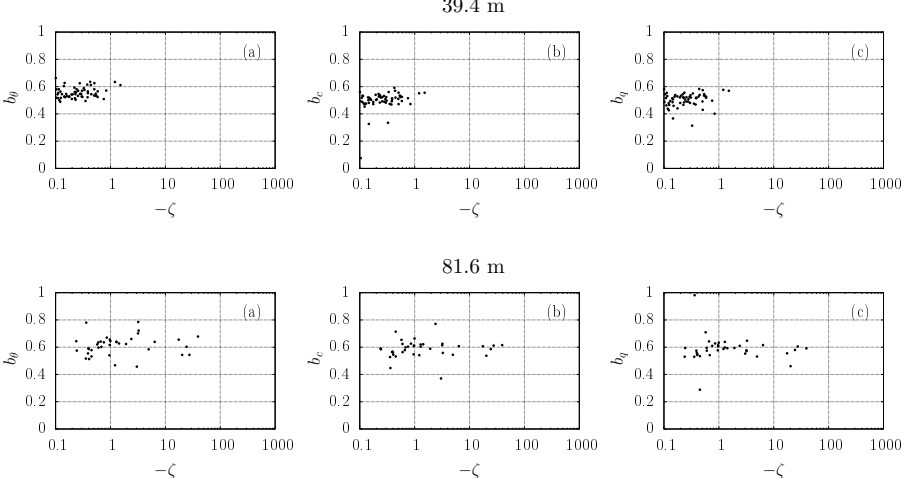

**Figure 11.** Relaxation coefficient variation with solar angle $- 0° < |Z| \leq 20°$. (a) and (d) are temperature, (b) and (e) are $CO_2$ and (c) and (f) are water vapor.

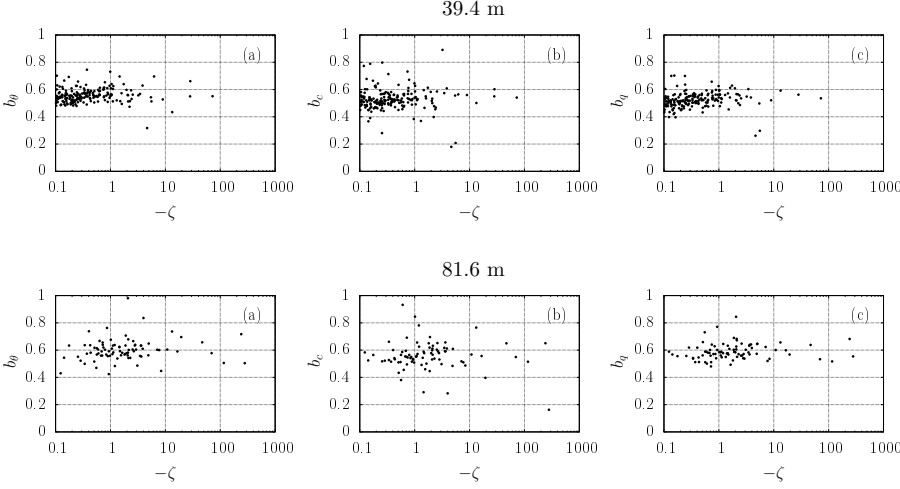

**Figure 12.** Relaxation coefficient variation with solar angle $- 20° < |Z| \leq 60°$. (a) and (d) are temperature, (b) and (e) are $CO_2$ and (c) and (f) are water vapor. Some data points were off the scale

affect the $\phi_w(\zeta)$ function, here, where the mean wind speed $\overline{u}$ is involved, we again detect a signal of $Z$-dependence.

The plots of $b_u$ against the stability variable $\zeta$ and zenith angle classes are shown in Fig. 14. Although in Table 5 it appears that the zenith angle is associated with the increase in scatter, in Fig.

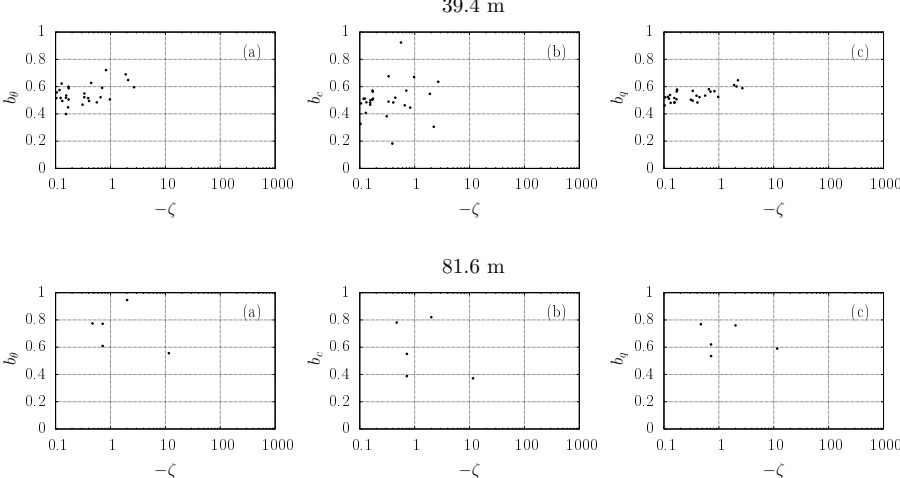

**Figure 13.** Relaxation coefficient variation with solar angle – $60° < |Z| \le 90°$. (a) and (d) are temperature, (b) and (e) are $CO_2$ and (c) and (f) are water vapor. Some data points were off the scale

**Table 5.** Statistics for $b_u$ (from REA) at 39.4 m.

|  | all data | $0° < |Z| \le 20°$ | $20° < |Z| \le 60°$ | $60° < |Z| \le 90°$ |
|---|---|---|---|---|
| Mean | 0.544 | 0.539 | 0.548 | 0.539 |
| Median | 0.531 | 0.524 | 0.531 | 0.536 |
| Standard deviation | 0.178 | 0.054 | 0.213 | 0.053 |

14 it is also possible that stability may play a role in the scatter as well. Clearly, this issue cannot be
completely decided with this dataset, and will require further investigation.

## 7   Conclusions

An experimental study of the behavior of scalars in the roughness sublayer has been made, with the objective of assessing their departure from the predictions of MOST.

The TKE dissipation rate departures are larger at the 39.4-m level and smaller at the 81.6-m level,
suggesting a gradual transition out of the roughness sublayer. This is not confirmed, however, by all turbulence statistics that we analyzed. For example, the dimensionless scalar standard deviations ($\phi_{\theta,q,c}$) at 39.4 and 81.6 do not show significant differences. $\phi_w$, on the other hand, remains much closer to the predictions of MOST at the two levels.

Moreover, an analysis of the scalar dissipation rates did not reveal any improvement in scalar be-
havior at smaller (ı.e. inertial-subrange) scales, indicating that the observed departures from MOST are occurring at all scales.

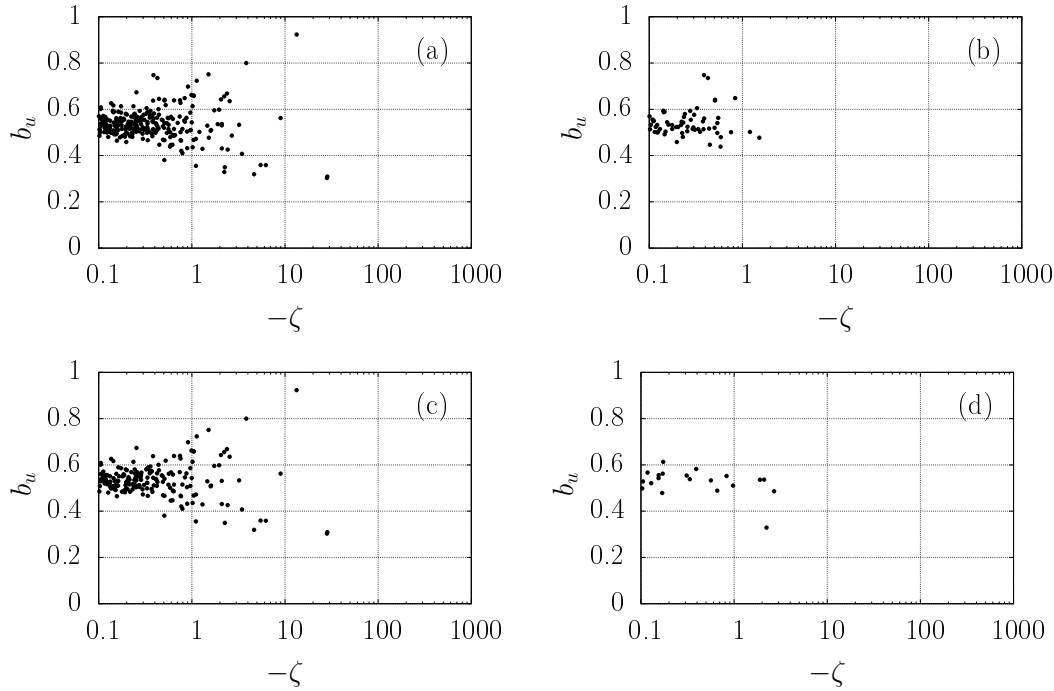

**Figure 14.** The behavior of $b_u$ at 39.4 m against Obukhov's stability variable $\zeta$ and zenith angle $Z$ in the RSL: (a) all data; (b) $0° < |Z| \leq 20°$; (c) $20° < |Z| \leq 60°$ ; and (d) $60° < |Z| \leq 90°$.

A significant finding in this work is that the degree of departure from MOST predictions is related to the zenith angle. This was found to impact several MOST functions, like $\phi_{\theta,q,c}$, rte, ste and the $b$ coefficient of the REA method, with significant less scatter than what is typically reported in the RSL for small zenith angles. Therefore, exploring these situations could lead to better flux estimates in the RSL. These results are strikingly similar to those found by Detto et al. (2010) with a storage term playing a similar role to $Z$, and possible links should be revisited in the future.

Fairly good adherence of $\phi_{\theta,q,c}$ to MOST (comparable to what is found in measurements above the RSL) was found for $0° < |Z| \leq 20°$, with increasingly larger scatter as $|Z|$ increases. It is possible that this is related to strongly dissimilar scalar sources and sinks in the vertical when the sun is low, when different physical (*i.e. heating*) and biophysical (*i.e. transpiration and photossynthesis*) forcings are produced throughout the canopy. It is also possible that, at these higher zenital angles, shading by the irregular tree heights produces a patchwork of different heat (and possibly water vapor and $CO_2$) sources, adding horizontal inhomogeneity.

The same pattern observed for the dimensionless scalar standard deviations appears with regard to the flux-similarity indices rte and ste. Again, they are closer to the theoretical values of $\pm 1$ and $+1$ (respectively) at lower zenith angles, agreeing with the classical MOST predictions.

Finally, the $b$ coefficients associated with the Relaxed Eddy Accumulation method were also affected by the zenith angle, with considerable improvement in the the range $0° < |Z| \leq 20°$. We confirmed that $b$ does not depend on stability $\zeta$ (for unstable conditions), and our values are in the same range as previously observed values. The value of $b$ close to the canopy (39.4 m) may turn out to be slightly lower than above (81.6 m), similarly to what is reported by Gao (1995): a possible variation with height within the RSL needs further research.

**Data Availability**

All data used in this study are kept in the ATTO Database at *Instituto de Pesquisas da Amazônia*. Access should be requested to the ATTO Project Leaders.

*Acknowledgements.* We thank the Max Planck Society and the Instituto Nacional de Pesquisas da Amazonia for continuous support. We acknowledge the support by the German Federal Ministry of Education and Research (BMBF contract 01LB1001A) and the Brazilian Ministério da Ciência, Tecnologia e Inovação (MCTI/FINEP contract 01.11.01248.00) as well as the Amazon State University (UEA), FAPEAM, LBA/INPA and SDS/CEUC/RDS-Uatumã. Einara Zahn thanks Brazil's CAPES for her Master's scholarship. Leonardo Sá, Antônio Manzi and Nelson L. Dias thank Brazil's National Research & Technology Development Council (CNPq) for their "Productivity in Research" Grants 303728/2010-8, 312431/2013-9 and 303581/2013-1.

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
