# Peer review of "Scalar turbulent behavior in the roughness sublayer of an Amazonian forest"

_Atmospheric Chemistry and Physics, 2015_

## Referee Comment (RC1) · Anonymous Referee #1 · 18 Feb 2016

Review of ACP-2015-1043 "Scalar turbulent behavior in the roughness sublayer of an Amazonian forest" by Einara Zahn et al.

The study by Zahn et al. focuses on the analysis of the atmospheric boundary layer structure in the roughness sublayer of an Amazonian forest. Measurements of atmospheric turbulence made at several levels of the 82-m tower are used to study turbulent fluxes, scaling laws for turbulent mixing, and dissipation rates of various scalars (temperature, water vapour, and carbon dioxide). In this paper, the authors extensive debate on the breakdown of Monin-Obukov similarity theory (MOST) in the roughness sublayer under unstable conditions. I understand that the authors have done an extensive data analysis and reported among other things the importance of the solar zenith angle for similarity of scalars. In general, the authors have set out the problem and carefully worked out what needs to be done to address several issues. The paper is

original, makes a significant contribution and is well written. I recommend acceptance of the paper for publication in the ACP with minor revisions, not so much in terms of redoing the analysis, but rather providing perspective on important questions and difficulties.

Specific Comments:

1. My main concern is associated with self-correlation (also referred to as artificial or spurious correlation), which occurs in some plots because of the shared variables. Awareness regarding the self-correlation has been increasing in the past several years (e.g., Andreas and Hicks, 2002; Klipp and Mahrt, 2004; Baas et al., 2006; Grachev et al. 2007 and papers surveyed therein). Authors say nothing about this problem. However, some of the results (e.g., plots of various similarity functions in Figures 4-7) may be suffered by self-correlation because have built-in correlation that is not associated with real physics. For example, increasing sigma_w/u* with increasing -z/L ('1/3' power law) is likely associated with self-correlation because same variables (friction velocity) appear in two quantities between which a functional relationship is sought. I would like to see here some discussion on this point.

2. Section 2.1 (page 4, line 111). Use of a CSAT3 sonic anemometer now requires flow distortion correction. The recent controversy concerning the underestimation of vertical wind speed by non-orthogonal sonic anemometers has largely been resolved (see papers by Horst et al. (2015, BLM - reference is below) and B51K-01 (Frank et al.) and B51K-02 (Horst et al.) at the 2014 American Geophysical Union Fall meeting). I recommend the authors download Tom Horst's AGU talk and BLM paper, available at his website http://www.eol.ucar.edu/homes/horst/ However, this likely would not affect the general results of this study.

3. Since the sonic anemometer measures the so-called 'sonic' virtual temperature (which is close to the virtual temperature) the moisture correction in the sonic anemometer signal is necessary to obtain the correct value of temperature itself and
sensible heat flux (e.g. Kaimal and Finnigan, 1994). Authors reported the standard deviation of the temperature (Figures 4-6) and temperature spectra (Eq.6). To value the present results the authors should either show that the moisture corrections and their impact on the results are small, or (if otherwise) apply moisture corrections to the sonic temperature following Schotanus et al. (1983) based on the data collected by LI-7500.

4. Authors say nothing about the Webb correction (called WPL or Webb effect after the paper by Webb et al. [1980]). This correction must be taken into account when the turbulent fluxes of minor constituents such as carbon dioxide or, in some cases, water vapor are measured (Webb et al. 1980).

5. Section 2.2. Important discussion on the QC recommendations by Klipp and Mahrt (2004) and Sanz Rodrigo and Anderson (2013, their Table 1) have been missed. I think the authors should also include these papers in their discussion.

Editorial/Technical Comments:

Page 6, line 164. Turbulence Kinetic Energy (TKE) has already been defined earlier in Section 3.1, line 136.

Section 5.1. Define the scalar's turbulent scales $a_*$, $c_*$ etc. Because, $c_* < 0$, the similarity functions for 'c' in Figs. 4-6 should be defined as $\sigma_c/|c_*|$.

Literature, not mentioned in the manuscript:

Andreas E.L, Hicks B.B. (2002) Comments on "Critical test of the validity of Monin-Obukhov similarity during convective conditions." J. Atmos. Sci. 59, 2605–2607.

Baas P., Steeneveld G.J., van de Wiel B.J.H., Holtslag A.A.M. (2006) Exploring Self-Correlation in Flux-Gradient Relationships for Stably Stratified Conditions. J. Atmos. Sci. 63(11), 3045–3054.

Grachev A.A., Andreas E.L, Fairall C.W., Guest P.S., Persson P.O.G. (2007) SHEBA

flux-profile relationships in the stable atmospheric boundary layer. Boundary-Layer Meteorol. 124(3), 315–333. DOI: 10.1007/s10546-007-9177-6

Horst T.W., Semmer S.R., Maclean G. (2015) Correction of a non-orthogonal, three-component sonic anemometer for flow distortion by transducer shadowing. Boundary-Layer Meteorology, 155(3): 371-395. DOI: 10.1007/s10546-015-0010-3

Klipp C.L., Mahrt L. (2004) Flux-Gradient Relationship, Self-correlation and Intermittency in the Stable Boundary Layer. Quart. J. Roy. Meteorol. Soc. 130(601), 2087–2103.

Sanz Rodrigo J, Anderson P.S. (2013) Investigation of the stable atmospheric boundary layer at Halley Antarctica. Boundary-Layer Meteorol. 148(3): 517-539. DOI: 10.1007/s10546-013-9831-0

Schotanus P., Nieuwstadt F.T.M., De Bruin H.A.R. (1983) Temperature measurement with a sonic anemometer and its application to heat and moisture fluxes. Boundary-Layer Meteorol. 26(1): 81–93. DOI: 10.1007/BF00164332

Webb E.K., Pearman G.I., Leuning R. (1980) Correction of flux measurements for density effects due to heat and water vapour transfer. Q. J. R. Meteorol. Soc. 106(447): 85–100. DOI: 10.1002/qj.49710644707
* * *

---

## Editor Comment (EC1) · G. Fisch (Editor) · 8 May 2016

**Review of Zahn et al. (2016) – "Scalar turbulent behavior in the roughness sublayer**

**of an Amazonian forest".**

General comment: The work reports new data in the roughness sublayer (RSL) above tall forests and features them in relation to Monin-Obukhov similarity theory (MOST). The main result of interest is that when the zenith angle is low, the radiation load appears to force higher spatial uniformity thereby making the flow resemble surface layers (and hence follow MOST) – at least for heat and some of the biologically active scalars such as CO2 and water vapor. All in all, the data are unique, and the analysis opens up new ways to thinking about the RSL. For these reasons, the paper may be published in Atmos. Chem. Phys. The comments below are mainly for the authors to consider – and should be viewed as lines of improvement.

1.  The choice of variables analyzed ($\sigma$'s, velocity skewness, $\phi_\epsilon$, the temperature variance dissipation, and b) have not been justified in an integrated manner. Perhaps the authors meant to state that some of the variables are used to identify whether the flow statistics are in the RSL or not – and some variables are relevant to the stated goal of analyzing VOC measurements using flux-gradient relations and REA. May be structuring the rationale along those lines upfront is worthwhile. That is, the work will be dealing with variables that describe the turbulent Schmidt (and Prandtl) numbers and eddy diffusivity for momentum in the RSL – as well as the similarity in *b* across scalars (and momentum) in the RSL.

2.  The linkage between equations (8) and (9) is not entirely clear. The excursions represented by $c'$ are not synonymous to $\overline{c^+} - \overline{c^-}$. I think the authors can do a much better job at justifying the high-order velocity and scalar statistics to *b*.

3.  A follow-up on comment 2, since this work is all about simulations to determine *b*, it is worth comparing *b* for scalars and momentum – that is

$$\overline{w'u'} = b_u \sigma_w \left( \overline{u'^+} - \overline{u'^-} \right).$$

Whether $b_u/b_s$ (where s is a scalar) is constant for various zenith angles and stability conditions is worth reporting (as $b_u$ may be far more sensitive to the roughness elements here than the source-sink distribution).

4.  The equality in coefficients $b_s$ does not necessarily require perfect similarity. For example, for two scalars say s1 and s2, then $R_{w,s1} \frac{\sigma_{s1}}{\left( \overline{s1'^+} - \overline{s1'^-} \right)} = b_{s1}$, and $R_{w,s2} \frac{\sigma_{s2}}{\left( \overline{s2'^+} - \overline{s2'^-} \right)} = b_{s2}$. Equating $b_{s1}$ to $b_{s2}$ does not necessarily require that $R_{w,s1} = R_{w,s2}$ as should be evident from the aforementioned definitions.

5.  Horizontal and vertical velocity skewness values – the flume experiments by Poggi et al. (2004) – figure below suggests that the cross-over height of skewness sign reversal dependence on the

vegetation density per se. For dense canopies, the figure below suggests that both skewness values switch signs – and at different levels. This hints that the definition of the RSL thickness will vary with the statistic being analyzed (as expected).

[Figure]

*Figure 3.* Variation of temporally- and horizontally-averaged moments with normalized height ($z/h$) for (a) mean longitudinal velocity, (b) mean shear stress, (c) longitudinal velocity standard deviation ($\sigma_u = \overline{u'^2}^{1/2}/u_*$), (d) vertical velocity standard deviation ($\sigma_w = \overline{w'^2}^{1/2}/u_*$), (e-f) longitudinal and vertical velocity skewness ($sk_u = \overline{u'^3}/\sigma_u^3$ and $sk_w = \overline{w'^3}/\sigma_w^3$), and (g-h) longitudinal and vertical velocity kurtosis ($ku_u = \overline{u'^4}/\sigma_u^4$ and $ku_w = \overline{w'^4}/\sigma_w^4$). Solid lines represent the sparsest and densest canopies.

6. The apparent agreement between $\frac{\sigma_w}{u_*}$ and MOST scaling may be due to self-correlation (see Cava et al., 2008). Certainly, more needs to be done to make a convincing case it is not all about self-correlation.

7. The authors should comment that all the normalized variances are above MOST predictions – but as discussed in Katul et al. (1995), inhomogeneity in the RSL impacts variances (i.e. the variance exceeds what would have predicted by the flux alone) but not necessarily fluxes. So, why is this result significant to VOC measurements – the fluxes may be the same but the variances higher in the RSL? Unless the authors meant to tie this finding to their REA and similarity theories (i.e. to quantities such as $\frac{\sigma_{s1}}{\left(\overline{s1'^+}-\overline{s1'^-}\right)}$).

8. A corollary comment to point 7 - Why did the authors focus only on the unstable conditions? Stable conditions (i.e. cooling) are equally important to shed light on the de-activation here, especially for heat and $CO_2$ (sources and sinks switch signs) may be worth exploring.

9. Pages 14-15, the role of storage may be significant (see Detto et al.,2010). Also, the authors may want to inspect Figure 7 in Detto et al. (2008).

10. The conclusions need to present a revised picture of the RSL – does this mean production and dissipation of TKE and temperature variance does not hold – and if so – what does that imply to the usage of K-theory or even the interpretation of constant fluxes with height? More important, the authors should attempt to explore $\frac{dF_s}{dz} \neq 0$ and its relation to zenith angle? Or $g_\theta(\zeta)$ or $\chi$? This has the most practical consequence of whether fluxes are constant with height or not.

References:

Detto et al., 2008, *Agricultural and Forest Meteorology*, 14, 902-916.

Detto et al., 2010, *Boundary-Layer Meteorology*, 136, 407-430.

Cava et al., 2008, *Boundary Layer Meteorology*, 128, 33-57.

Katul et al., 1995, *Boundary Layer Meteorology*, 74, 237-260.

Poggi et al. 2004, *Boundary-Layer Meteorology*, 111, 565-587.

---

## Referee Comment (RC2) · Anonymous Referee #2 · 9 May 2016

This is a very important study, well presented by the authors. It has two main qualities: - Usefulness: with the upcoming tall tower in the Amazon, a study of this kind is essential. It will serve as an important reference for future flux analysis at the site; - An important new insight: the finding that zenith angle "controls" the validity of the similarity hypothesis is, to my knowledge, a new one. The reasoning proposed by the authors to explain it makes sense.

For these reasons, I recommend publication. I have a few suggestions to the authors, so I am rating it as "accept pending minor revisions":

1. After all screening and flitering for data quality and stationarity, a small percentage of the total data is left for the analysis. Although I understand that this is part of the nature of turbulence data, I feel it would be important to openly discuss how this fact

affects the significance of the results. Do the authors expect that a similar percentage of flux measurements will also need to be discarded at the site? Mabe the authors could mention that when using MOST to infer fluxes from profiles, a larger fraction of data may be used. I also suggest you show (in a table,maybe) what fraction of data is used for each range of zenith angle considered in the analysis. That would hint the reader whether the problems associated with high zenith angle are the same ones that cause lack of stationarity, for instance.

2. In section 3.2, you associate intermittency to skewness. Although I can understand the idea, it sounds strange, because there is another statistical momment, kurtosis, that is inherently associated with intermittency. The same association is later done at line 168.

3. line 181. The concern about similarity validity depending on the large scales of the flow is certainly a valid one, but it had never been presented before, in the manuscript. Given its importance, I expect that some readers will read the introduction with that idea in mind. I suggest addressing it before.

4. Figure 1 is not very informative. How about presenting histograms for each level?

5. Why aren't Sk_u values shown?, given tht they are even included in the discussion that precedes the figure?

335 The $u$-skewness values are now given: see lines 273–283, and Figure of the current version of the manuscript.

6. Likewise, in figure 3, the dissipation rates (epsilons) could be directly shown as a function of z/L, along with the existent similarity expressions for this relationship. I would be very curious to see that.

7. line 329. I like the explanation for the similarity between CO2 and temperature, based on the signs of entrainment. I would also add that the surface fluxes between these two quantities also have opposite signals in the vast majority of the cases.

small issues:

line 40: remove "to" at the end of the line; line 239: "and" instead of "e"

---

## Author Comment (AC1) · 30 Jun 2016

We are answering all questions and comments by 3 referees in the enclosed pdf file. We are also attaching a revised manuscript.

Please also note the supplement to this comment: http://www.atmos-chem-phys-discuss.net/acp-2015-1043/acp-2015-1043-AC1-supplement.zip

---

## Author Response (AR1)

Manuscript prepared for Atmos. Chem. Phys.
with version 2015/11/06 7.99 Copernicus papers of the LaTeX class copernicus.cls.
Date: 16 July 2016

**Scalar turbulent behavior in the roughness sublayer of an Amazonian forest. Answer to Reviewers**

Zahn, Dias et al.

*Correspondence to:* Nelson L. Dias (`nldias@ufpr.br`)

**1 Answer to Editor's Letter:**

We thank the Editor and the Reviewers for the comments. We answer the all remaining comments and suggestions below.

**Reviewer(s)' Comments to Author:**

5 **Reviewer: 1**

Comments to the Author

The study by Zahn et al. focuses on the analysis of the atmospheric boundary layer structure in the roughness sublayer of an Amazonian forest. Measurements of atmospheric turbulence made at several levels of the 82-m tower are used to study turbulent fluxes, scaling laws for turbulent mixing, and dissipation rates of various scalars (temperature, water vapour, and carbon dioxide). In this paper,
10 the authors extensive debate on the breakdown of Monin-Obukov similarity theory (MOST) in the roughness sublayer under unstable conditions. I understand that the authors have done an extensive data analysis and reported among other things the importance of the solar zenith angle for similarity of scalars. In general, the authors have set out the problem and carefully worked out what needs to be done to address several issues. The paper is original, makes a significant contribution and is well
15 written. I recommend acceptance of the paper for publication in the ACP with minor revisions, not so much in terms of redoing the analysis, but rather providing perspective on important questions and difficulties.

We thank the reviewer for all his comments. We give a detailed answer to the specific below

20 1. My main concern is associated with self-correlation (also referred to as artificial or spurious correlation), which occurs in some plots because of the shared variables. Awareness regarding the self-correlation has been increasing in the past several years (e.g., Andreas and Hicks, 2002; Klipp and Mahrt, 2004; Baas et al., 2006; Grachev et al. 2007 and papers surveyed therein). Authors say nothing about this problem. However, some of the results (e.g., plots of

various similarity functions in Figures 4-7) may be suffered by self-correlation because have built-in correlation that is not associated with real physics. For example, increasing $\sigma_w/u*$ with increasing $-z/L$ (1/3 power law) is likely associated with self-correlation because same variables (friction velocity) appear in two quantities between which a functional relationship is sought. I would like to see here some discussion on this point.

We thank the reviewer for bringing up the issue of self-correlation, which is something we had not addressed in the previous version. The issue is complex, and undoubtedly still to some degree controversial since, as acknowledged from the start, the possibility that self-correlation contaminate data analysis does not by itself contradict the general validity of similarity theories (see, for example Hicks, 1981; Johansson et al., 2002). A very effective way to test the extent to which self-correlation contaminates the analysis is still contained in the suggestion by Andreas and Hicks (2002) of randomizing the data set. Examples of application of this "self-correlation test" for the $\phi$'s can be found in von Randow et al. (2006) and Cava et al. (2008).

Following (Cava et al., 2008), our original plots were comparared with the plots produced with randomised $u_*$ (we retained the same scalar variances and covariances). The results can be seen in the figure below, which depicts the original plots of Figure 4 in the manuscript against the same data randomized (in red), where the dashed lines show a $-1/3$ power law. Our results are similar to Cava et al. (2008)'s results: although the randomised data follows the $-1/3$ power law, it erroneously follows this tendency even at near-neutral regime, where the original data follows the predicted similarity functions. In this case, as stated by Cava et al. (2008), the correlations are not "spurious" in the near-neutral regimes.

We also calculated the scalar fluxes for fairly to highly unstable conditions ($-\zeta > 0.2$) using the flux-variance method, and compared them with the measured fluxes. We obtained high correlations (in fact, higher than those reported by Cava et al. (2008)). As these authors comment, these fluxes are calculated without knowledge of $u_*$, and the success of the method under these more unstable conditions gives independent confirmation that spurious correlation is not contaminating our analyses.

[Figure]

Next, the same results are displayed for each zenith angle range.

[Figure]

For completeness, we do discuss self-correlation in lines 304–317 of the current version of the manuscript.

2. Section 2.1 (page 4, line 111). Use of a CSAT3 sonic anemometer now requires flow distortion correction. The recent controversy concerning the underestimation of vertical wind speed by non-orthogonal sonic anemometers has largely been resolved (see papers by Horst et al. (2015, BLM - reference is below) and B51K-01 (Frank et al.) and B51K-02 (Horst et al.) at the 2014 American Geophysical Union Fall meeting). I recommend the authors download Tom Horst's AGU talk and BLM paper, available at his website www.eol.ucar.edu/homes/horst/ However, this likely would not affect the general results of this study.

We thank the reviewer for the helpful suggestion. In this case, as the reviewer correctly predicted, flow distortion corrections do not change appreciably any of our results. To show this,

we did apply the corrections suggested by Horst et al. (2015). to the CSAT-3 data. As an example, the figure below compares the $u'$ fluctuations before and after the corrections. As can be seen, the differences are only minor.

70

[Figure]

Next, we also checked the effects of flow distortion on some of our statistics (in this case, $\sigma_w/u_*$). The results with and without correction are shown in the figure below.

[Figure]

To summarize, after careful consideration, flow distortion corrections do not change our results appreciably. With this in mind, we chose not to apply them in the results shown in the paper. At any rate, we mention this issue explicitly in the paper now, in lines 132–137.

3. Since the sonic anemometer measures the so-called sonic virtual temperature (which is close to the virtual temperature) the moisture correction in the sonic anemometer signal is necessary to obtain the correct value of temperature itself and sensible heat flux (e.g. Kaimal and Finnigan, 1994). Authors reported the standard deviation of the temperature (Figures 4-6) and temperature spectra (Eq.6). To value the present results the authors should either show that the moisture corrections and their impact on the results are small, or (if otherwise) apply moisture corrections to the sonic temperature following Schotanus et al. (1983) based on the data collected by LI-7500.

We agree with the Reviewer. We note that the Schotanus et al. paper actually deals with corrections of the sonic temperature which are already incoporated into the outputs of both the CSAT3 and the Gill sonic anemometers according to the sonics' manuals. Regarding the data themselves, we have re-calculated all temperature, humidity and $CO_2$ data to derive good estimates of instantaneous thermodynamic temperature, specific humidity and $CO_2$ mass concentration. With these data, no further density corrections are needed, and the calculation of the

turbulence statistics presented in the paper are straightforward. Details about this procedure are now given in lines 124–131 of the current version of the manuscript.

95     We would also like to add that the actual changes in our former results are very small. As an example, we present a comparison for temperature $\sigma_\theta/\theta_*$ statistics in the figure below: black means former results and red means corrected values.

[Figure]

100    4. Authors say nothing about the Webb correction (called WPL or Webb effect after the paper by Webb et al. [1980]). This correction must be taken into account when the turbulent fluxes of minor constituents such as carbon dioxide or, in some cases, water vapor are measured (Webb et al. 1980).

We agree. We have re-calculated all our data making the necessary corrections. This is already 105     mentioned in the answer above. Again, the detailed descriptions and references now appear in lines 124–131 of the current version of the manuscript.

5. Section 2.2. Important discussion on the QC recommendations by Klipp and Mahrt (2004) and Sanz Rodrigo and Anderson (2013, their Table 1) have been missed. I think the authors should also include these papers in their discussion.

We thank the reviewer for suggesting the additional QC in the aforementioned papers. After careful consideration, we came to the conclusion that these additional criteria cannot impact our data: they deal with problems typical of stable conditions (mainly low-intensity or nonexisting turbulence), whereas we only worked with unstable data. Moreover, we did exclude all runs whose two-minute standard deviations were less than a specified threshold (see lines 141–144 of the current version). Incidentally, our own current research on the topic has shown that this test of ours is quite capable of identifying those low-intensity turbulence runs in stable conditions as well (Zahn et al., 2016). In view of that, we decided not to include these references.

6. Page 6, line 164. Turbulence Kinetic Energy (TKE) has already been defined earlier in Section 3.1, line 136.

   Done in line 196

7. Section 5.1. Define the scalar's turbulent scales $a_*$, $c_*$ etc. Because, $c_* < 0$, the similarity functions for 'c' in Figs. 4-6 should be defined as $\sigma_c/|c_*|$.

   Done, in the new equation 9 and in lines 299–303

8. Literature, not mentioned in the manuscript

   (a) Andreas E.L, Hicks B.B. (2002) Comments on "Critical test of the validity of Monin-Obukhov similarity during convective conditions." J. Atmos. Sci. 59, 2605–2607.

   (b) Baas P., Steeneveld G.J., van de Wiel B.J.H., Holtslag A.A.M. (2006) Exploring Self-Correlation in Flux-Gradient Relationships for Stably Stratified Conditions. J. Atmos. Sci. 63(11), 3045–3054.

   (c) Grachev A.A., Andreas E.L, Fairall C.W., Guest P.S., Persson P.O.G. (2007) SHEBA flux-profile relationships in the stable atmospheric boundary layer. Boundary-Layer Meteorol. 124(3), 315–333. DOI: 10.1007/s10546-007-9177-6

   (d) Horst T.W., Semmer S.R., Maclean G. (2015) Correction of a non-orthogonal, three-component sonic anemometer for flow distortion by transducer shadowing. Boundary-Layer Meteorology, 155(3): 371-395. DOI: 10.1007/s10546-015-0010-3

   (e) Klipp C.L., Mahrt L. (2004) Flux-Gradient Relationship, Self-correlation and Intermittency in the Stable Boundary Layer. Quart. J. Roy. Meteorol. Soc. 130(601), 2087–2103.

   (f) Sanz Rodrigo J, Anderson P.S. (2013) Investigation of the stable atmospheric bound- ary layer at Halley Antarctica. Boundary-Layer Meteorol. 148(3): 517-539. DOI: 10.1007/s10546-013-9831-0

   (g) Schotanus P., Nieuwstadt F.T.M., De Bruin H.A.R. (1983) Temperature measurement with a sonic anemometer and its application to heat and moisture fluxes. Boundary-Layer Meteorol. 26(1): 81–93. DOI: 10.1007/BF00164332

145     (h) Webb E.K., Pearman G.I., Leuning R. (1980) Correction of flux measurements for density effects due to heat and water vapour transfer. Q. J. R. Meteorol. Soc. 106(447): 85–100. DOI: 10.1002/qj.49710644707

We thank the reviewer for the helpful suggestions. Many of the above works have now been incorporated into the references of the current version of the manuscript.

150 **Reviewer: 2**

Comments to the Author

The work reports new data in the roughness sublayer (RSL) above tall forests and features them in relation to Monin-Obukhov similarity theory (MOST). The main result of interest is that when the zenith angle is low, the radiation load appears to force higher spatial uniformity thereby making the
155 flow resemble surface layers (and hence follow MOST) - at least for heat and some of the biologically active scalars such as CO2 and water vapor. All in all, the data are unique, and the analysis opens up new ways to thinking about the RSL. For these reasons, the paper may be published in Atmos. Chem. Phys. The comments below are mainly for the authors to consider - and should be viewed as lines of improvement.
160     We thank the reviewer for all his comments and the corresponding improvments that (we believe) resulted in the manuscript. We give a detailed answer to the comments in this review below

1. The choice of variables analyzed (sigma's, velocity skewness, $\phi_\epsilon$, the temperature variance dissipation, and $b$) have not been justified in an integrated manner. Perhaps the authors meant to state that some of the variables are used to identify whether the flow statistics are in the RSL
165     or not – and some variables are relevant to the stated goal of analyzing VOC measurements using flux-gradient relations and REA. May be structuring the rationale along those lines upfront is worthwhile. That is, the work will be dealing with variables that describe the turbulent Schmidt (and Prandtl) numbers and eddy diffusivity for momentum in the RSL - as well as the similarity in b across scalars (and momentum) in the RSL.

170     We thank the reviewer for the suggestion. It is incomporated in lines 89–95 of the current version of the manuscript

2. The linkage between equations (8) and (9) is not entirely clear. The excursions represented by c' are not synonymous to $\overline{c^+} - \overline{c^-}$. I think the authors can do a much better job at justifying the high-order velocity and scalar statistics to b. We agree with the reviewer: the equations
175     for the original Eddy Accumulation method were confusing because we don't apply the EA, and because the Relaxed Eddy Accumulation method equation does not follow from them. Therefore, they were eliminated from the present version: see lines 225–231 of the current version.

3. A follow-up on comment 2, since this work is all about simulations to determine b, it is worth comparing b for scalars and momentum. Whether $b_u/b_s$( s is a scalar) is constant for various zenith angles and stability conditions is worth reporting (as $b_u$ may be far more sensitive to the roughness elements here than the sourcesink distribution).

We thank the reviewer for the comment. In this regard, we introduced new results on $b_u$ at the end of section 6, lines 435–450. Overall, there are indications that the zenith angle $Z$ may also influence $b_u$, but they are not as strong as with $b_c$, $b_\theta$ and $b_q$. We chose not to present results in terms of $b_u/b_s$: To keep the paper's narrative cohesive, results are given in terms of $b_u$, as done with the scalars. However, we do show in the next four figures below the results for $b_u/b_s$ against $\zeta$ for the totality of data and for the three zenith-angle classes. We believe that the results incorporated into the manuscript are easier to interpret and deal with the subject to the extent that is possible with the present dataset.

all data:

[Figure]

$0 < |Z| \le 20$:

[Figure]

$20 < |Z| \le 60$:

[Figure]

$60 < |Z| \leq 90$

4. The equality in coefficients $b_s$ does not necessarily require perfect similarity. For example, for two scalars say $s_1$ and $s_2$, then $R_{w,s_1} \frac{\sigma_{s1}}{\overline{s_1^{+\prime}} - \overline{s_1^{-\prime}}} = b_{s_1}$, and $R_{w,s_2} \frac{\sigma_{s2}}{\overline{s_2^{+\prime}} - \overline{s_2^{-\prime}}} = b_{s_2}$. Equating $b_{s_1}$ to $b_{s_2}$ does not necessarily require that $R_{w,s_1} = R_{w,s_2}$ as should be evident from the aforementioned definitions.

The reviewer is of course right, and we admit the oversight. We have changed the text accordingly in lines 246–248.

5. Horizontal and vertical velocity skewness values - the flume experiments by Poggi et al. (2004) - figure below suggests that the cross-over height of skewness sign reversal dependence on the vegetation density per se. For dense canopies, the figure below suggests that both skewness values switch signs – and at different levels. This hints that the definition of the RSL thickness will vary with the statistic being analyzed (as expected).

[Figure]

*Figure 3.* Variation of temporally- and horizontally-averaged moments with normalized height ($z/h$) for (a) mean longitudinal velocity, (b) mean shear stress, (c) longitudinal velocity standard deviation ($\sigma_u = \overline{u'^2}^{1/2}/u_*$), (d) vertical velocity standard deviation ($\sigma_w = \overline{w'^2}^{1/2}/u_*$), (e-f) longitudinal and vertical velocity skewness ($sk_u = \overline{u'^3}/\sigma_u^3$ and $sk_w = \overline{w'^3}/\sigma_w^3$), and (g-h) longitudinal and vertical velocity kurtosis ($ku_u = \overline{u'^4}/\sigma_u^4$ and $ku_w = \overline{w'^4}/\sigma_w^4$). Solid lines represent the sparsest and densest canopies.

We thank the reviewer for pointing this out. First, we included the new reference in the discussion of $Sk_u$ in the (new) lines 189–192. Then, we included results for the $u$-skewness in the new Figure 5, and changed the text accordingly in lines 273–283. Overall, the new $Sk_u$ analysis confirms the results previously obtained with $Sk_w$.

6. The apparent agreement between $\sigma_w/u^*$ and MOST scaling may be due to self-correlation (see Cava et al., 2008). Certainly, more needs to be done to make a convincing case it is not all about selfcorrelation.

Please see our answer to the same issue in item 1, Reviewer 1, above. Like Caval et al., we did randomize our dataset and like them we concluded that our results do not suffer from spurious correlation.

7. The authors should comment that all the normalized variances are above MOST predictions - but as discussed in Katul et al. (1995), inhomogeneity in the RSL impacts variances (i.e. the variance exceeds what would have predicted by the flux alone) but not necessarily fluxes. So, why is this result significant to VOC measurements – the fluxes may be the same but the variances higher in the RSL? Unless the authors meant to tie this finding to their REA and similarity theories (i.e. to quantities such as $\frac{\sigma_{s1}}{s_1^{+'} - s_1^{-'}}$.)

We discuss the point further, following the reviewer's suggestion, in lines 331–334. But the crucial point is the following: in this manuscript, we are investigating fruitful approaches to

230  understand the Amazonian RSL a little better, and maybe to better estimate fluxes. It would take us too far (and out of scope) to actually devise and test methodologies for flux estimation procedures applicable to substances such as VOCs whose high-frequency measurement is difficult. Notice that if all we have is a flux-variance behavior with stability of the type shown in Fig. 4, and if the flux-variance method is the only one available, then the corresponding flux

235  estimates will suffer from the large scatter in the data. Then, if we now look at Fig. 6 (small zenith angles), the scatter is much less: if the flux-variance method were the only one available, fluxes estimated for small zenith angles would be more reliable. Of course, this is still being verified with high-frequency data, but the important fact here (and one that we believe has not been reported yet) is that there are at least some situations where the difficulties as-

240  sociated with the roughness sublayer could be somewhat alleviated. We found similar results with the REA, and this is promising, because unlike the flux-variance method it dispenses with high-frequency measurements. The important point, therefore, is to open up these alternatives for flux estimates. This is now reinforced in lines 462–467.

8. A corollary comment to point 7 - Why did the authors focus only on the unstable conditions?

245  Stable conditions (i.e. cooling) are equally important to shed light on the de-activation here, especially for heat and CO2 (sources and sinks switch signs) may be worth exploring.

We fully agree. Some of our colleagues in the ATTO project (L. Sá and O. Acevedo and collaborators) are exploring stable conditions in depth. On the other hand, most of the mass transfer (in terms of the time integral of the fluxes) still takes place in daytime conditions, and

250  we realized that it would be important to contribute to better daytime flux estimates to begin with. It is also common to analyze stable and unstable conditions separately. In this work, we decided to focus exclusively (in this current version) on unstable conditions. We hope to explore this issue from the stable side in the future.

9. Pages 14-15, the role of storage may be significant (see Detto et al.,2010). Also, the authors

255  may want to inspect Figure 7 in Detto et al. (2008).

We thank the reviewer for bringing up this point. We were not aware of Detto et al.'s (2010) results, and they are important to mention. Indeed, it is clear that the results for the storage term found by Detto et al. (2010) are similar to our results for the zenith angle in the sense that the storage term is larger at high zenith angles, and that better agrement was found by

260  Detto et al. for low storage (in the middle of the day, when the zenith angles are lower in absolute value). We included this discussion in the manuscript in lines 362–369. Unfortunately we don't have good profiles to test this, and this comparison couldn't be done by us at the moment. Also, we agree that Detto et al.'s (2008) quadrant analysis would be a powerful tool for better understanding the causes of scalar dissimilarity in the RSL. However, we believe

265  that this manuscript is already covering a lot of different aspects, so we decided not to include

such analyses at this point. At any rate, an explicit mention to this possibility is given in lines 386–397.

10. The conclusions need to present a revised picture of the RSL — does this mean production and dissipation of TKE and temperature variance does not hold — and if so — what does that imply to the usage of K-theory or even the interpretation of constant fluxes with height? More important, the authors should attempt to explore $dF/dz \neq 0$ and its relation to zenith angle? Or $g_\theta(\zeta)$ or $\chi$. This has the most practical consequence of whether fluxes are constant with height or not.

The reviewer touches on crucial questions, and it would be an honour to be able to really provide a revised picture of the RSL. At this point, however, we will need to keep to humbler objectives. The reasons are as follows: turbulent budgets need reliable mean profile data, and for this preliminary field campaign we do not have simultaneous good profile data (as already explained above). We also could not find a systematic behavior regarding either TKE or scalar dissipation, as our discussion on $g_\theta$ shows. Therefore, we cordially defer more definitive conclusions on the Amazonian RSL for the time when better and more comprehensive data are collected by ATTO.

[revised manuscript text omitted]